# FADRM: Fast and Accurate Data Residual Matching for Dataset Distillation

**Jiacheng Cui**[*1], **Xinyue Bi**[*2] , **Yaxin Luo**[1], **Xiaohan Zhao**[1], **Jiacheng Liu**[1], **Zhiqiang Shen**[†1]

[1]VILA Lab, MBZUAI    [2]University of Ottawa

[*]Equal Contribution    [†]Corresponding Author

**Code: FADRM (GitHub)**

## Abstract

Residual connection has been extensively studied and widely applied at the model architecture level. However, its potential in the more challenging data-centric approaches remains unexplored. In this work, we introduce the concept of ***Data Residual Matching*** for the first time, leveraging data-level skip connections to facilitate data generation and mitigate data information vanishing. This approach maintains a balance between newly acquired knowledge through pixel space optimization and existing core local information identification within raw data modalities, specifically for the dataset distillation task. Furthermore, by incorporating training-time refinements, our method significantly improves computational efficiency, achieving superior performance while reducing training time and peak GPU memory usage by 50%. Consequently, the proposed method **F**ast and **A**ccurate **D**ata **R**esidual **M**atching for Dataset Distillation (**FADRM**) establishes a new state-of-the-art, demonstrating substantial improvements over existing methods across multiple dataset benchmarks in both efficiency and effectiveness. For instance, with ResNet-18 as the student model and a 0.8% compression ratio on ImageNet-1K, the method achieves 48.4% test accuracy in single-model dataset distillation and 50.9% in multi-model dataset distillation, surpassing RDED by +6.4% and outperforming state-of-the-art multi-model approaches, EDC and CV-DD, by +2.3% and +4.9%.

## 1 Introduction

In recent years, the computer vision and natural language processing communities have predominantly focused on model-centric research, driving an unprecedented expansion in the scale of neural networks. Landmark developments such as LLMs and MLLMs in ChatGPT [26, 1], Gemini [36], DeepSeek [19] and other large-scale foundation models have shown the tremendous potential of deep learning architectures. However, as these models grow in complexity, the dependency on high-quality, richly informative datasets has become increasingly apparent, setting the stage for a paradigm shift towards data-centric approaches. Historically, the emphasis on building bigger and more complex models has often overshadowed the critical importance of the data. While model-centric strategies have

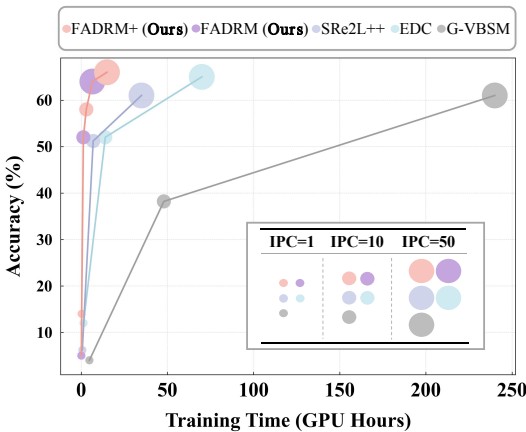

Figure 1: Total training hours on a single RTX-4090 *vs.* test set accuracy, comparing prior state-of-the-art methods with our proposed framework (+ denotes multi-model distillation).

39th Conference on Neural Information Processing Systems (NeurIPS 2025).

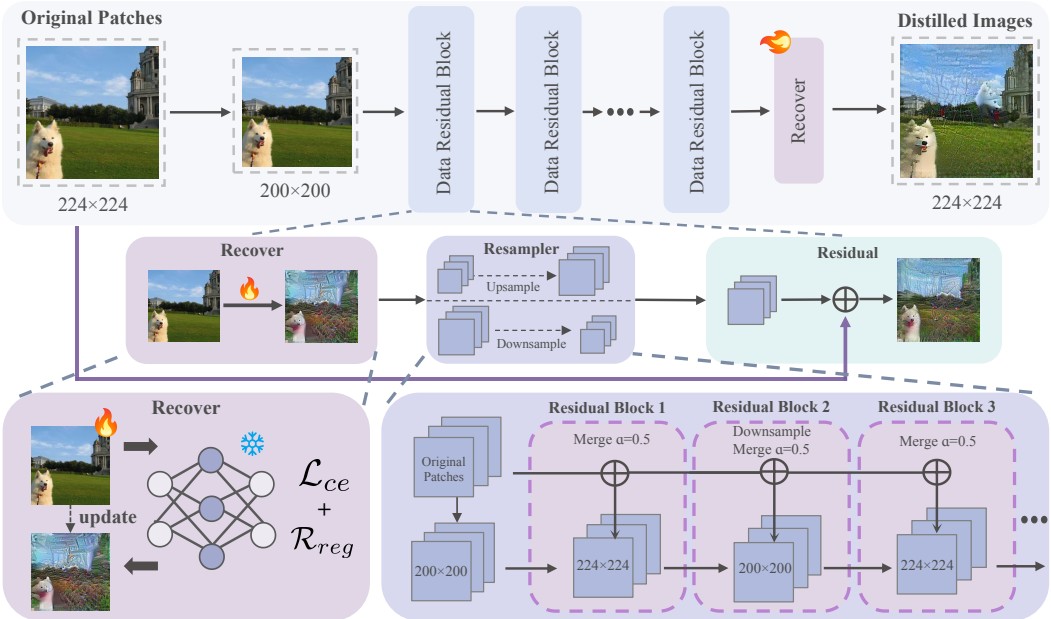

Figure 2: Overview of **FADRM**. It starts by downsampling the real data patches (both $1\times1$ and $2\times2$ [33] can be used as initialization and perform well in our experiments, meanwhile imposing downsampling to reduce cost). These downsampled images are subsequently processed through a series of proposed *Data Residual Blocks*. Each block utilizes a pretrained model to optimize the images within a predefined optimization budget, resamples them to a target resolution, and incorporates residual connections from the original patches via a mixing ratio $\alpha$. Finally, the images undergo an additional recovery stage, without residual connections, to produce the final distilled data.

delivered impressive results, they tend to overlook the benefits of optimizing data quality, which is essential for achieving higher performance with lower data demands. Recent advancements in data-centric research highlight the importance of improving information density, reducing the volume of required data, and expediting the training process of large-scale models, thus presenting a more holistic approach to performance enhancement.

Within this evolving landscape, dataset distillation [39], also called dataset condensation [14, 47, 43] has emerged as a pivotal area of research. The goal of dataset distillation is to compress large-scale datasets into smaller, highly informative subsets that retain the essential characteristics of the original data. This approach not only accelerates the training process of complex models but also mitigates the storage and computational challenges associated with massive datasets. Despite significant progress, many existing state-of-the-art methods in dataset distillation still struggle with issues related to scalability, generalization across diverse data resolutions, realism and robustness.

While residual connections have been well studied and widely implemented in the model architecture design field, primarily to prevent gradient vanishing and ensure effective feature propagation, their potential within data-centric paradigms remains largely unexplored. At the model level, residual connections help maintain the flow of gradients and enable deeper network architectures. In contrast, at the data level, similar connections can potentially prevent the loss of critical original dataset information and improve scalability and generalization across architectures during the data distillation process. This observation and design introduce a novel perspective on leveraging residual mechanisms beyond traditional model optimization, especially in the challenging domain of dataset optimization.

In this work, we introduce for the first time the concept of *Data Residual Matching* for dataset distillation. Our approach leverages data-level skip connections, a novel idea for data-centric task to prevent real data information vanishing in multi-block data synthesis architecture. We call our method **F**ast and **A**ccurate **D**ata **R**esidual **M**atching (**FADRM**), which, as shown in Fig. 2, employs a multi-resolution image recovery scheme that utilizes image resolution shrinkage and expansion in a residual manner, thereby capturing fine-grained details and facilitating the recovery of both global and local information. This balance between newly acquired knowledge through pixel space

optimization and the preservation of existing core local information within raw data modalities marks a significant advancement in dataset distillation. By integrating these data-level residual connections, our approach enhances the generalization and robustness of the distilled datasets.

Exhaustive empirical evaluations of our proposed **FADRM** on CIFAR-100 [15], Tiny-ImageNet [16], ImageNet-1K [8] and its subset demonstrate that it not only accelerates the dataset distillation process by 50% but also achieves superior accuracy that beats all previous state-of-the-art methods on both accuracy and generation speed. This approach effectively **bridges the gap between model-centric and data-centric paradigms**, providing a robust solution to the challenges inherent in high-quality data generation. Our contributions in this paper are as follows:

- We extend conventional residual connection from the model level to the data level area, and present for the first time a simple yet effective, theoretically grounded residual connection design for data generation to enhance data-centric task.

- We introduce a novel dataset distillation framework based on the proposed *data residual matching*, incorporating multi-scale residual connections in data synthesis to improve both efficiency and accuracy.

- Our approach achieves state-of-the-art results across multiple datasets, such as CIFAR-100, Tiny-ImageNet and ImageNet-1K, while being more efficient and requiring less computational cost than all previous methods.

## 2  Related Work

**Dataset Distillation** aims to synthesize a compact dataset that retains the critical information of a larger original dataset, enabling efficient training while maintaining performance comparable to the full dataset. Overall, the matching criteria include *Meta-Model Matching* [39, 25, 23, 49, 9, 12], *Gradient Matching* [47, 45, 18, 14, 48], *Trajectory Matching* [4, 7, 5, 10], Distribution Matching [46, 38, 21, 17, 27, 32, 41], and *Uni-level Global Statistics Matching* [43, 30, 31, 42, 6, 40]. Dataset distillation on large-scale datasets has recently attracted significant attention from the community. For a detailed overview, it can be referred to the newest survey works [29, 20] on this topic.

**Efficient Dataset Distillation.** Several methods improve the computational efficiency of dataset distillation. TESLA [7] accelerates MTT [4] via batched gradient computation, avoiding full graph storage and scaling to large datasets. DM [46] sidesteps bi-level optimization by directly matching feature distributions. SRe²L [43] adopts a Uni-Level Framework that aligns synthetic data with pretrained model statistics. G-VBSM [30] extends this by using lightweight model ensembles. EDC [31] further boosts efficiency through real data initialization, accelerating convergence.

**Residual Connection in Network Design.** Residual connections have played a pivotal role in advancing deep learning. Introduced in ResNet [11] to alleviate vanishing gradients, they enabled deeper networks by improving gradient flow. This idea was extended in Inception-ResNet [34] through multi-scale feature integration, and further generalized in DenseNet [13] via dense connectivity and feature reuse. Residual designs have also been central to Transformer architectures [37].

## 3  Approach

**Preliminaries.** Let the original dataset be denoted by $\mathcal{O} = \{(x_i, y_i)\}_{i=1}^{|\mathcal{O}|}$, and let the goal of *dataset distillation* be to construct a compact synthetic dataset $\mathcal{C} = \{(\tilde{x}_j, \tilde{y}_j)\}_{j=1}^{|\mathcal{C}|}$, with $|\mathcal{C}| \ll |\mathcal{O}|$, such that the model $f_{\theta_\mathcal{C}}$ trained on $\mathcal{C}$ exhibits similar generalization behavior to the model $f_{\theta_\mathcal{O}}$ trained on $\mathcal{O}$. This objective can be formulated as minimizing the performance gap over the real data distribution:

$$\arg\min_{\mathcal{C}, |\mathcal{C}|} \sup_{(x,y) \sim \mathcal{O}} |\mathcal{L}\left(f_{\theta_\mathcal{O}}(x), y\right) - \mathcal{L}\left(f_{\theta_\mathcal{C}}(x), y\right)| \tag{1}$$

where the parameters $\theta_\mathcal{O}$ and $\theta_\mathcal{C}$ are obtained via empirical risk minimization:

$$\theta_\mathcal{O} = \arg\min_\theta \mathbb{E}_{(x,y) \sim \mathcal{O}}[\mathcal{L}(f_\theta(x), y)], \quad \theta_\mathcal{C} = \arg\min_\theta \mathbb{E}_{(\tilde{x},\tilde{y}) \sim \mathcal{C}}[\mathcal{L}(f_\theta(\tilde{x}), \tilde{y})]. \tag{2}$$

The goal is to generate $\mathcal{C}$ in order to maximize model performance with minimal data. Among existing methods, a notable class directly optimizes synthetic data without access to the original dataset,

referred to as *uni-level optimization*. While effective, this approach faces two key limitations: (1) progressive information loss during optimization, termed *information vanishing*, and (2) substantial computational and memory costs for large-scale synthesis, limiting real-world applicability.

**Information Vanishing.** In contrast to images distilled using bi-level frameworks, the information content in images generated by uni-level methods (e.g., EDC [31]) is fundamentally upper-bounded, as the original dataset is not utilized during synthesis (see Theorem 1). As optimization progresses, the information density initially increases but eventually deteriorates due to the accumulation of local feature loss. This degradation leads to information vanishing (see Fig. 3), which significantly reduces the fidelity of the distilled images and limits their effectiveness in downstream tasks.

**Theorem 1** (Proof in Appendix A.2). *Let $f_\theta$ denote a pretrained neural network on the original dataset $\mathcal{O}$, with fixed parameters and corresponding BatchNorm statistics $\mathbf{BN}^{\mathrm{RM}}$ and $\mathbf{BN}^{\mathrm{RV}}$. Let $\tilde{x}$ be a synthetic sample obtained by minimizing a general objective $\mathcal{L}(f_\theta; x)$ that relies exclusively on information extracted from the pretrained model $f_\theta$ (e.g., its predictive output distribution or internal feature statistics alignment), that is, $\tilde{x} = \arg\min_x \mathcal{L}(f_\theta; x)$. Define*

$$H(f_\theta) = \sup_{x \in \mathrm{supp}(\mathcal{O})} H(f_\theta(x)) \tag{3}$$

*as the maximum per-sample Shannon entropy of the model's output. Then, the mutual information between the optimized distilled dataset $\mathcal{C} = \{(\tilde{x}_j, \tilde{y}_j)\}_{j=1}^{|\mathcal{C}|}$ and the original dataset $\mathcal{O}$ is bounded by*

$$I(\mathcal{C}; \mathcal{O}) \leq |\mathcal{C}| \, H(f_\theta). \tag{4}$$

The insight of this theorem is that if the pretrained model $f_\theta$ is overly confident on all inputs (low maximum entropy), then $H(f_\theta)$ is small, and thus the distilled set, no matter how we optimize it, cannot encode a large amount of information about $\mathcal{O}$.

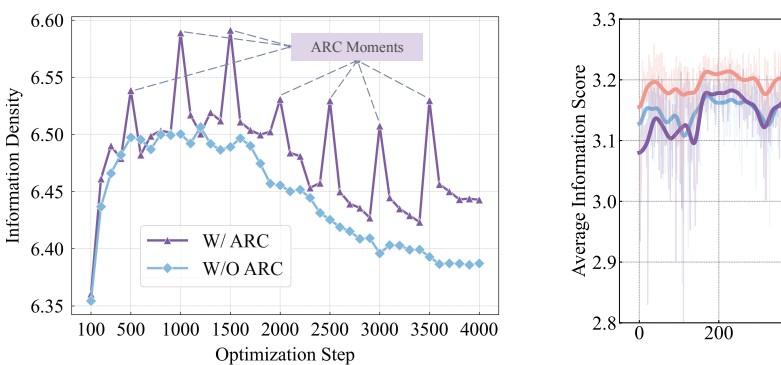 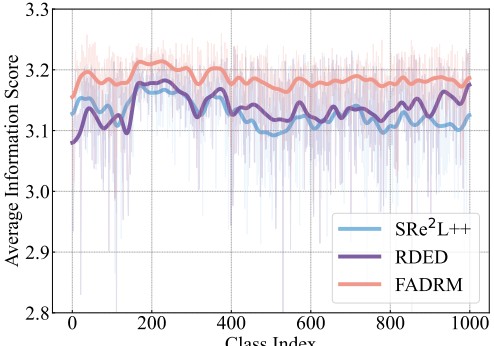

Figure 3: The above figures illustrate the phenomenon of *Information Vanishing*. The **Left** Figure shows the evolution of information density across optimization steps, quantified through feature-level entropy using a pretrained ResNet-18 [11], comparing uni-level optimization (W/O *ARC*) with our **FADRM** (W/ *ARC*). The gray lines highlight the information density enhancement achieved through residual connection. The **Right** Figure shows the comparison of information scores (higher is better) across different classes, measured by *pixel-level entropy*, among **FADRM**, SRe$^2$L++, and RDED. All experiments are conducted on a distilled ImageNet-1K dataset with IPC=10.

**Computational Challenges.** Although uni-level frameworks exhibit scalability to large-scale datasets, the overall time required to generate a large distilled dataset remains prohibitively expensive. As illustrated in Fig. 1, EDC [31] requires nearly 70 hours to generate a 50 IPC distilled dataset, which limits its applicability in contexts involving repeated runs, large-scale data synthesis, or comprehensive empirical analysis. This motivates the need for more computationally efficient optimization strategies.

## 3.1 Overview of FADRM

The proposed **FADRM** framework, as illustrated in Fig. 2 and detailed in Algorithm 1, addresses the limitations of existing uni-level optimization frameworks by integrating three proposed components: (1) *MPT*: a mixed-precision training scheme that accelerates optimization and reduces computation

by casting model parameters to lower-precision formats, (2) *MRO*: a multi-resolution optimization that improves computational efficiency, and (3) *ARC*: an adjustable embedded residual mechanism designed to seamlessly integrate essential features from the original dataset. This framework ensures both efficiency and generation fidelity in the optimization process.

---

**Algorithm 1 FADRM**: Residual Matching for Dataset Distillation

---

**Require:** Recover Model $f_\theta$, Total Training Iters $N_{\text{iter}}$, Real Patches $\mathbf{P}_s$, Merge Ratio $\alpha$, Downsampled Resolutions $D_{\text{ds}}$, Original Resolutions $D_{\text{orig}}$, Number of *ARC*s $k$
**Ensure:** Distilled image $\tilde{x}_{N_{\text{iter}}}$
1:  $n_{\text{iter}} \leftarrow \lfloor N_{\text{iter}}/(k+1) \rfloor, \quad \tilde{x}_0 \leftarrow \text{RESAMPLE}(\mathbf{P}_s, D_{\text{ds}})$
2:  **for** $i = 1$ **to** $k$ **do**
3:      **for** $t = 1$ **to** $n_{\text{iter}}$ **do**
4:          $\tilde{x}_{(i-1)n_{\text{iter}}+t} \leftarrow \text{GRADSTEP}(f_\theta, \tilde{x}_{(i-1)n_{\text{iter}}+t-1}) \triangleright$ Optimize $\tilde{x}$ to align the property of $f_\theta$
5:      **end for**
6:      $\tilde{x}_{in_{\text{iter}}} \leftarrow \begin{cases} \text{RESAMPLE}(\tilde{x}_{in_{\text{iter}}}, D_{\text{orig}}), & \text{if } \texttt{Shape}(\tilde{x}_{in_{\text{iter}}}) = D_{\text{ds}} \\ \text{RESAMPLE}(\tilde{x}_{in_{\text{iter}}}, D_{\text{ds}}), & \text{otherwise} \end{cases}$
7:      $\tilde{x}_{in_{\text{iter}}} \leftarrow \alpha \tilde{x}_{in_{\text{iter}}} + (1-\alpha) \cdot \text{RESAMPLE}(\mathbf{P}_s, \texttt{Shape}(\tilde{x}_{in_{\text{iter}}}))$
8:  **end for**
9:  **for** $t = 1$ **to** $N_{\text{iter}} - kn_{\text{iter}}$ **do**
10:     $\tilde{x}_{kn_{\text{iter}}+t} \leftarrow \text{GRADSTEP}(f_\theta, \tilde{x}_{kn_{\text{iter}}+t-1})$
11: **end for**
12: **return** $\tilde{x}_{N_{\text{iter}}}$

---

## 3.2 Mixed Precision Training for Data Generation

Previous uni-level frameworks typically retain a fixed training pipeline, seeking efficiency through architectural or initialization-level changes. In contrast, we explicitly optimize the training process by incorporating Mixed Precision Training (MPT) [24]. Specifically, we convert the model parameters $\theta$ from FP32 to FP16 and utilize FP16 for both logits computation and cross-entropy loss evaluation. To preserve numerical stability and ensure accurate distribution matching, we retain the computation of the divergence to the global statistics (Appendix D), as well as the gradients of the total loss with respect to $\tilde{x}$ in FP32. By integrating *MPT*, our framework significantly reduces both GPU memory consumption and training time by approximately 50%, thereby significantly enhancing efficiency.

## 3.3 Multi-resolution Optimization

Multi-Resolution Optimization (*MRO*) enhances computational efficiency by optimizing images across multiple resolutions, unlike conventional methods that operate on a fixed input size. Naturally, low-resolution inputs can reduce computational cost for the model, they often come at the expense of performance. To mitigate this, our method periodically increases the data resolution back at specific stages, resulting in a mixed-resolution optimization process, as illustrated in Fig. 2 (bottom-right). This approach is particularly beneficial for large-scale datasets (e.g., ImageNet-1K), where direct high-resolution optimization is computationally inefficient. Notably, optimization time scales significantly with input size for large datasets but remains stable for smaller ones (input size $\leq 64$). Thus, *MRO* is applied exclusively to large-scale datasets, as downscaling offers no efficiency gains for smaller ones. Specifically, given an initialized image $\mathbf{P}_s \in \mathbb{R}^{D_{\text{orig}} \times D_{\text{orig}} \times C}$, we first downsample it into a predefined resolution $D_{\text{ds}}$ utilizing bilinear interpolation (detailed in Appendix E):

$$\tilde{x}_0 = \text{Resample}(\mathbf{P}_s, D_{\text{ds}}), \quad D_{\text{ds}} < D_{\text{orig}} \tag{5}$$

The downscaled images $\tilde{x}_0$ are optimized over $n_{\text{iter}} = \lfloor N_{\text{iter}}/(k+1) \rfloor$ iterations, yielding the refined result $\tilde{x}_{n_{\text{iter}}}$. Subsequently, $\tilde{x}_{n_{\text{iter}}}$ is upscaled to its original resolution,

$$\tilde{x}_{n_{\text{iter}}} = \text{Resample}(\tilde{x}_{n_{\text{iter}}}, D_{\text{orig}}) \tag{6}$$

The upscaled image $\tilde{x}_{n_{\text{iter}}}$ is further optimized within the same budget $n_{\text{iter}}$ to recover information lost during the downscaling and upscaling processes. This iterative procedure (downscaling optimization

and upscaling optimization) is repeated until the total optimization iteration $N_{\text{iter}}$ is exhausted. To ensure *MRO* attains efficiency gains without quality loss, selecting an appropriate $D_{\text{ds}}$ is crucial. Too small a $D_{\text{ds}}$ causes information loss, while too large yields minimal efficiency improvement. Hence, $D_{\text{ds}}$ should be carefully balanced for efficiency and effectiveness.

**Saved Computation by *MRO*.** Assume the forward computation cost scales as $\mathcal{O}(D^2 C)$. The baseline method performs all $N_{\text{iter}}$ steps at full resolution $D_{\text{orig}}$, yielding: $\text{Cost}_{\text{baseline}} = N_{\text{iter}} \cdot \mathcal{O}(D_{\text{orig}}^2 C)$. **FADRM** performs $k$ alternating-resolution stages of $n_{\text{iter}} = \lfloor N_{\text{iter}}/(k+1) \rfloor$ steps, with approximately half at downsampled resolution $D_{\text{ds}}$. Let $r = (D_{\text{ds}}/D_{\text{orig}})^2$. The cost ratio is:

$$\frac{\text{Cost}_{MRO}}{\text{Cost}_{\text{baseline}}} = 1 - \frac{n_{\text{iter}}}{N_{\text{iter}}} \cdot \left( \left\lceil \tfrac{k}{2} \right\rceil (1 - r) \right) \tag{7}$$

Under fixed $N_{\text{iter}}$ and $k$, the cost ratio decreases linearly with $1 - r$. Smaller $r$ (i.e., more aggressive downsampling) yields greater savings, but may compromise data fidelity.

## 3.4 Adjustable Residual Connection

In uni-level optimization, the absence of the original dataset leads to information vanishing which significantly degrades the feature representation of the distilled dataset. To mitigate this issue, we introduce Adjustable Residual Connection (*ARC*), a core mechanism that mitigates information vanishing (see Fig. 3) and improves the robustness of the distilled data (see Theorem 2 and Theorem 3). Essentially, *ARC* iteratively fuses the intermediate optimized image $\tilde{x}_t \in \mathbb{R}^{D_t \times D_t \times C}$ at iteration $t$ with the resized initialized data patches $\tilde{\mathbf{P}}_t$, which contain subtle details from the original dataset. Formally, the update rule is defined as:

$$\tilde{x}_t = \alpha \tilde{x}_t + (1 - \alpha)\text{Resample}(\mathbf{P}_s, D_t) \tag{8}$$

where $\alpha \in [0, 1]$ is a tunable merge ratio gov-erning the contribution of original dataset infor-

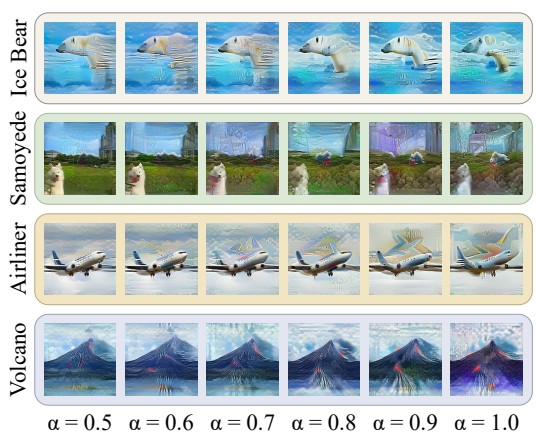

$\alpha = 0.5 \quad \alpha = 0.6 \quad \alpha = 0.7 \quad \alpha = 0.8 \quad \alpha = 0.9 \quad \alpha = 1.0$

Figure 4: Visualization of the distilled images with varying merge ratios using **FADRM**.

mation. A smaller $\alpha$ strengthens the integration of details from $\mathbf{P}_s$, whereas a larger $\alpha$ prioritizes the preservation of the global features in the $\tilde{x}_t$. This trend is visualized in Fig. 4. *ARC* introduces a hyperparameter $k$, which determines the frequency of residual injections. Given a total optimization budget of $N_{\text{iter}}$, the training process is divided into $k + 1$ segments, where residual connections occur after every $n_{\text{iter}} = \lfloor N_{\text{iter}}/(k + 1) \rfloor$ iterations. The update follows:

$$\tilde{\mathbf{P}}_{in_{\text{iter}}} = \text{Resample}\left(\mathbf{P}_s, D_{in_{\text{iter}}}\right), \quad \tilde{x}_{in_{\text{iter}}} = \alpha \tilde{x}_{in_{\text{iter}}} + (1 - \alpha)\tilde{\mathbf{P}}_{in_{\text{iter}}}. \tag{9}$$

where $i \in \{1, 2, \dots, k\}$ denotes the index of the residual injection stage, and $D_{in_{\text{iter}}}$ indicates the spatial resolution of the intermediate image at the corresponding iteration $t = in_{\text{iter}}$. The final phase involves pure optimization without further residual injections. Notably, *ARC* conducts per-element weighted fusion of two image tensors with negligible overhead. With a complexity of $\mathcal{O}(H_t W_t C)$, it scales linearly with pixel and channel counts, making it suitable for high-resolution data.

**Theorem 2** (Proof in Appendix A.3.1). *Let $\mathcal{H}$ be a class of functions $h : \mathbb{R}^d \to \mathbb{R}$, and let $h$ be Lipschitz-continuous with constant $L_h > 0$, and the loss function $\ell$ be Lipschitz-continuous with constant $L_l > 0$ and bounded within a finite range $[0, B]$. Consider: 1. Optimized perturbation added to the original data: $\tilde{\mathcal{C}}^{res} = \{\tilde{x}_i^{res}, \tilde{y}_i^{res}\}_{i=1}^n$. 2. residual injected dataset (FADRM): $\tilde{\mathcal{C}}_{\text{FADRM}} = \{\tilde{x}_i, \tilde{y}_i\}_{i=1}^n$. 3. patches selected from the original dataset: $\mathcal{O} = \{x_i, y_i\}_{i=1}^n$. 4. discrepancy $\Delta := \frac{1}{n}\sum_{i=1}^n \|\tilde{x}_i^{res} - x_i\|$. Let $h_{\text{res}} \in \mathcal{H}$ denote the hypothesis trained on $\tilde{\mathcal{C}}^{res}$, and $h_{\text{FADRM}} \in \mathcal{H}$ be trained on $\tilde{\mathcal{C}}_{\text{FADRM}}$. Define the corresponding empirical risks: $\widehat{\mathcal{L}}_{\text{res}} := \frac{1}{n}\sum_{i=1}^n \ell(h_{\text{res}}(\tilde{x}_i^{res}), \tilde{y}_i^{res})$, $\widehat{\mathcal{L}}_{\text{FADRM}} := \frac{1}{n}\sum_{i=1}^n \ell(h_{\text{FADRM}}(\tilde{x}_i), \tilde{y}_i)$. Assume:*

$$\mathfrak{R}_n(\mathcal{H} \circ \mathcal{O}) < \mathfrak{R}_n(\mathcal{H} \circ \tilde{\mathcal{C}}^{res}) \tag{10}$$

*Then the generalization bound of $h_{\mathrm{FADRM}}$ is rigorously shown to be tighter than that of $h_{\mathrm{res}}$, i.e.,*

$$\widehat{\mathcal{L}}_{\mathrm{FADRM}} + 2B \cdot \mathfrak{R}_n(\mathcal{H} \circ \tilde{\mathcal{C}}_{\mathrm{FADRM}}) < \widehat{\mathcal{L}}_{\mathrm{res}} + 2B \cdot \mathfrak{R}_n(\mathcal{H} \circ \tilde{\mathcal{C}}^{res}). \tag{11}$$

The key insight of this theorem is that when synthetic data is heavily optimized, thereby increasing the complexity of the hypothesis space and making the model more susceptible to overfitting, combining it with the more structured original data can yield a tighter generalization bound. We further demonstrate the advantage of **FADRM** over the Uni-level framework through Theorem 3.

**Theorem 3** (Proof in Appendix A.3.2). *Let $\hat{x}_{uni}$ denote the image generated by the uni-level framework, and $\hat{x}_{fadrm}$ denote the image generated by FADRM using both $f_\theta$ and real image patches $\mathcal{O}$. Then the mutual information between the generated image and the original data satisfies,*

$$I(\hat{x}_{fadrm}; \mathcal{O}) > I(\hat{x}_{uni}; \mathcal{O}).$$

This theorem shows that **FADRM**'s improvement primarily arises from its direct access to the original dataset during optimization, thereby enhancing the amount of captured information.

# 4 Experiments

## 4.1 Datasets and Experimental Setup

**Datasets.** We conduct experiments across datasets with varying resolutions, including CIFAR-100 (32×32) [15], Tiny-ImageNet (64×64) [16], ImageNet-1K (224×224) [8], and their subsets.

**Baseline Methods.** To evaluate the effectiveness of our proposed framework, we conduct a comprehensive comparison against three state-of-the-art dataset distillation baselines. The first baseline is RDED [33], which selects cropped patches directly from the original dataset and is therefore categorized as involving full participation of the original data. The second method, EDC [31], retains a high degree of original data participation by optimizing selected patches with an extremely small learning rate, producing synthetic images that are close to the original samples. The third method, CV-DD [6], aligns global BatchNorm statistics with sufficient optimization by updating initialization, resulting in minimal original data involvement despite initialization from real patches. These baselines exhibit varying degrees of original data involvement, providing a solid basis for evaluating **FADRM**.

## 4.2 Main Results

**Results Analysis.** As shown in Table 2, our framework consistently achieves state-of-the-art performance across various settings. For instance, on ImageNet-1K with IPC=10 and ResNet-101 as the student model, the ensemble-enhanced variant **FADRM+** attains an accuracy of 58.1%, outperforming EDC and CV-DD by a substantial margin of +6.4%. Notably, RDED underperforms **FADRM**, underscoring the limitations of relying solely on the original dataset without further optimization. Furthermore, CV-DD is inferior to **FADRM+**, highlighting the drawbacks of largely excluding original data during synthesis. Lastly, the consistent outperformance of **FADRM+** over EDC validates the efficacy of our framework in harnessing original data via data-level residual connections.

**Information Vanishing.** To quantify the issue of information vanishing, we report the generalization accuracy of models trained on distilled images generated by **FADRM** (w/ and w/o *ARC*) across different optimization steps. As shown in Table 1, prolonged optimization without *ARC* leads to accuracy degradation, indicating information loss. In contrast, applying *ARC* stabilizes the optimization process and maintains steady performance gains, highlighting its effectiveness in mitigating information vanishing and retaining essential representational cues.

| Iterations | w/ ARC (%) | w/o ARC (%) |
|---|---|---|
| 500 | 40.0 | 39.7 |
| 1000 | 45.2 | 44.8 |
| 1500 | 47.2 | 46.5 |
| 2000 | 47.7 | 46.1 |
| 2500 | 48.0 | 46.5 |
| 3000 | 48.0 | 45.4 |
| 3500 | 48.2 | 44.5 |
| 4000 | 48.4 | 43.1 |

Table 1: Generalization accuracy (%) across varying optimization steps.

**Efficiency Comparison.** Table 3 (Left) highlights the superior efficiency of our framework compared to existing Uni-level frameworks. Bi-level frameworks are excluded from this comparison due to their inherent limitations in scalability for large-scale datasets. Specifically, **FADRM+** achieves a reduction of 3.9 seconds per image in optimization time

| Dataset | IPC (Ratio) | ResNet-18 | | | | | ResNet-50 | | | | | ResNet-101 | | | | |
|---|---|---|---|---|---|---|---|---|---|---|---|---|---|---|---|---|
| | | RDED | EDC | CV-DD | FADRM | FADRM+ | RDED | EDC | CV-DD | FADRM | FADRM+ | RDED | EDC | CV-DD | FADRM | FADRM+ |
| CIFAR-100 | 1 (0.2%) | 17.1 | 39.7 | 28.3 | 31.8 | **40.6** | 10.9 | 36.1 | 28.7 | 27.3 | **37.4** | 11.2 | 32.3 | 29.0 | 29.2 | **40.1** |
| | 10 (2.0%) | 56.9 | 63.7 | 62.7 | 67.4 | **67.9** | 41.6 | 62.1 | 61.5 | 66.5 | **67.4** | 54.1 | 61.7 | 63.8 | 68.3 | **68.9** |
| | 50 (10.0%) | 66.8 | 68.6 | 67.1 | 71.0 | **71.3** | 64.0 | 69.4 | 68.2 | 71.5 | **72.1** | 67.9 | 68.5 | 67.6 | 71.9 | **72.1** |
| | Whole Dataset | | | 78.9 | | | | | 79.9 | | | | | 79.5 | | |
| Tiny-ImageNet | 1 (0.2%) | 11.8 | 39.2 | 30.6 | 28.6 | **40.4** | 8.2 | 35.9 | 25.1 | 28.4 | **39.4** | 9.6 | 40.6 | 28.0 | 27.9 | **41.9** |
| | 10 (2.0%) | 41.9 | 51.2 | 47.8 | 48.9 | **52.8** | 38.4 | 50.2 | 43.8 | 47.3 | **53.7** | 22.9 | 51.6 | 47.4 | 47.8 | **53.6** |
| | 50 (10.0%) | 58.2 | 57.2 | 54.1 | 56.4 | **58.7** | 45.6 | 58.8 | 54.7 | 57.0 | **60.3** | 41.2 | 58.6 | 54.1 | 57.2 | **60.8** |
| | Whole Dataset | | | 68.9 | | | | | 71.5 | | | | | 70.6 | | |
| ImageNette | 1 (0.1%) | 35.8 | - | 36.2 | 36.2 | **39.2** | 27.0 | - | 27.6 | 31.1 | **31.9** | 25.1 | - | 25.3 | 26.3 | **29.3** |
| | 10 (1.0%) | 61.4 | - | 64.1 | 64.8 | **69.0** | 55.0 | - | 61.4 | 64.1 | **68.1** | 54.0 | - | 61.0 | 61.9 | **63.7** |
| | 50 (5.2%) | 80.4 | - | 81.6 | 83.6 | **84.6** | 81.8 | - | 82.0 | 84.1 | **85.4** | 75.0 | - | 80.0 | 80.3 | **82.3** |
| | Whole Dataset | | | 93.8 | | | | | 89.8 | | | | | 89.3 | | |
| ImageWoof | 1 (0.1%) | 20.8 | - | 21.4 | 21.0 | **22.8** | 17.8 | - | 19.1 | 19.5 | **19.9** | 19.6 | - | 19.9 | 20.0 | **21.8** |
| | 10 (1.1%) | 38.5 | - | 49.3 | 44.5 | **57.3** | 35.2 | - | 47.8 | 44.9 | **54.1** | 31.3 | - | 42.6 | 40.4 | **51.4** |
| | 50 (5.3%) | 68.5 | - | 71.9 | 72.3 | **72.6** | 67.0 | - | 71.2 | 71.0 | **71.7** | 59.1 | - | 69.9 | 70.3 | **70.6** |
| | Whole Dataset | | | 88.2 | | | | | 77.8 | | | | | 82.7 | | |
| ImageNet-1K | 1 (0.1%) | 6.6 | 12.8 | 9.2 | 9.0 | **14.7** | 8.0 | 13.3 | 10.0 | 12.2 | **16.2** | 5.9 | 12.2 | 7.0 | 6.8 | **14.1** |
| | 10 (0.8%) | 42.0 | 48.6 | 46.0 | 48.4 | **50.9** | 49.7 | 54.1 | 51.3 | 54.5 | **57.5** | 48.3 | 51.7 | 51.7 | 54.8 | **58.1** |
| | 50 (3.9%) | 56.5 | 58.0 | 59.5 | 60.1 | **61.2** | 62.0 | 64.3 | 63.9 | 65.4 | **66.9** | 61.2 | 64.9 | 62.7 | 66.0 | **67.0** |
| | Whole Dataset | | | 72.3 | | | | | 78.6 | | | | | 79.8 | | |

Table 2: **Post-evaluation performance comparison with SOTA baseline methods.** All experiments follow the training settings established in EDC [31]: 300 epochs for Tiny-ImageNet (IPC=10, 50), ImageNet-1K, and its subsets, and 1,000 epochs for CIFAR-100, Tiny-ImageNet (IPC=1). For fair comparison with single-model distillation (RDED) and ensemble-based methods (CV-DD, EDC), we evaluate both the single-model version (**FADRM** only utilized ResNet-18 [11] for distillation) and the ensemble-enhanced version (**FADRM+**). This evaluation strategy ensures equitable benchmarking while maintaining methodological consistency across all experiments.

| Method | Time Cost (s) | Peak Memory (GB) |
|---|---|---|
| SRe$^2$L++ [6] | 2.52 | 5.3 |
| **FADRM** | **0.47** | **2.9** |
| G-VBSM [30] | 17.28 | 21.4 |
| CV-DD [6] | 8.20 | 23.4 |
| EDC [31] | 4.99 | 17.9 |
| **FADRM+** | **1.09** | **11.0** |

| Model | #Params | RDED | EDC | CV-DD | FADRM+ |
|---|---|---|---|---|---|
| ResNet-18 [11] | 11.7M | 42.0 | 48.6 | 46.0 | **50.9** |
| ResNet-50 [11] | 25.6M | 49.7 | 54.1 | 51.3 | **57.5** |
| ResNet-101 [11] | 44.5M | 48.3 | 51.7 | 51.7 | **58.1** |
| EfficientNet-B0 [35] | 39.6M | 42.8 | 51.1 | 43.2 | **51.9** |
| MobileNetV2 [28] | 3.4M | 34.4 | 45.0 | 39.0 | **45.5** |
| ShuffleNetV2-0.5x [44] | 1.4M | 19.6 | 29.8 | 27.4 | **30.2** |
| Swin-Tiny [22] | 28.0M | 29.2 | 38.3 | – | **39.1** |
| Wide ResNet-50-2 [11] | 68.9M | 50.0 | – | 53.9 | **59.1** |
| DenseNet121 [13] | 8.0M | 49.4 | – | 50.9 | **55.4** |
| DenseNet169 [13] | 14.2M | 50.9 | – | 53.6 | **58.5** |
| DenseNet201 [13] | 20.0M | 49.0 | – | 54.8 | **59.7** |

Table 3: **Left:** Efficiency comparison between various optimization-based methods and our approach when distilling ImageNet-1K. The time cost is measured in seconds, representing the duration required to generate a single image on a single RTX 4090 GPU. **Right:** Top-1 accuracy (%) on ImageNet-1K for cross-architecture generalization with IPC=10.

compared to EDC [31], culminating in a total computational saving of 54 hours when applied to the 50 IPC ImageNet-1K dataset. Similarly, **FADRM** demonstrates a 28.5 hours reduction in training time relative to SRe$^2$L++ for the same task. Additionally, our framework significantly reduces peak memory usage compared to other frameworks, enabling efficient dataset distillation even in resource-constrained scenarios. These results underscore the scalability and computational efficiency of our approach, which not only accelerates large-scale dataset distillation but also substantially lowers associated computational costs.

## 4.3 Ablation Study

**Impact of Patch Numbers for Initialization and Residuals.** To assess the effect of different patch configurations during both the initialization and residual injection stages, we conduct an ablation study, as shown in Table 4. The results suggest that both $1 \times 1$ and $2 \times 2$ patch settings are effective for generating distilled data. However, the $1 \times 1$ configuration consistently delivers the better overall performance, making it the preferred choice in practice.

| | IPC=10 | | IPC=50 | |
|---|---|---|---|---|
| | FADRM | FADRM+ | FADRM | FADRM+ |
| $1 \times 1$ | **48.4** | **50.9** | **60.1** | **61.2** |
| $2 \times 2$ | 47.7 | 50.0 | 59.8 | 60.1 |

Table 4: Comparison of student model (ResNet-18) generalization performance when trained on distilled datasets generated using $1 \times 1$ and $2 \times 2$ patch configurations during initialization and residual injection.

**Impact of Mixed Precision Training (MPT).** Our ablation study in Table 5 shows that MPT preserves distilled dataset quality while significantly reducing peak memory usage and improving optimization efficiency, making it an effective strategy for accelerating distillation.

| | FADRM | | FADRM+ | | SRe$^2$L++ | | G-VBSM | |
| | W/ MPT | W/O MPT | W/ MPT | W/O MPT | W/ MPT | W/O MPT | W/ MPT | W/O MPT |
|---|---|---|---|---|---|---|---|---|
| ResNet-18 (Student) | 47.7 % | 47.8 % | 50.0 % | 49.6 % | 43.1 % | 43.1 % | 30.5 % | 30.7 % |
| Efficiency | 0.26 ms | 0.63 ms | 0.58 ms | 0.96 ms | 0.26 ms | 0.63 ms | 2.65 ms | 4.32 ms |
| Peak GPU Memory | 2.9 GB | 5.3 GB | 11.0 GB | 23.0 GB | 2.9 GB | 5.3 GB | 11.8 GB | 21.4 GB |

Table 5: Comparison of model generalization performance, optimization efficiency (milliseconds per image per iteration, measured under 100 batch size and 224 as input size), and peak GPU memory usage with and without mixed precision training under ImageNet-1K IPC=10.

| Configuration | Accuracy (%) | Time Cost (s) |
|---|---|---|
| **FADRM** (W/O *ARC* + W/O *MRO*) | 46.4 | 0.52 |
| **FADRM** (W/O *ARC* + W/ *MRO*) | 46.2 | 0.47 |
| **FADRM** (W/ *ARC* ($\alpha = 0.9$) + W/ *MRO*) | 45.7 | 0.47 |
| **FADRM** (W/ *ARC* ($\alpha = 0.8$) + W/ *MRO*) | 46.4 | 0.47 |
| **FADRM** (W/ *ARC* ($\alpha = 0.7$) + W/ *MRO*) | 47.6 | 0.47 |
| **FADRM** (W/ *ARC* ($\alpha = 0.6$) + W/ *MRO*) | 47.3 | 0.47 |
| **FADRM** (W/ *ARC* ($\alpha = 0.5$) + W/ *MRO*) | **47.7** | 0.47 |
| **FADRM** (W/ *ARC* ($\alpha = 0.4$) + W/ *MRO*) | 47.4 | 0.47 |

| Configuration | Accuracy (%) | Time Cost (s) |
|---|---|---|
| **FADRM+** (W/O *ARC* + W/O *MRO*) | 48.7 | 1.16 |
| **FADRM+** (W/O *ARC* + W/ *MRO*) | 48.2 | 1.09 |
| **FADRM+** (W/ *ARC* ($\alpha = 0.9$) + W/ *MRO*) | 48.5 | 1.09 |
| **FADRM+** (W/ *ARC* ($\alpha = 0.8$) + W/ *MRO*) | 48.0 | 1.09 |
| **FADRM+** (W/ *ARC* ($\alpha = 0.7$) + W/ *MRO*) | 48.9 | 1.09 |
| **FADRM+** (W/ *ARC* ($\alpha = 0.6$) + W/ *MRO*) | 49.3 | 1.09 |
| **FADRM+** (W/ *ARC* ($\alpha = 0.5$) + W/ *MRO*) | **50.0** | 1.09 |
| **FADRM+** (W/ *ARC* ($\alpha = 0.4$) + W/ *MRO*) | 49.5 | 1.09 |

Table 6: Performance comparison of ResNet-18 as the student model trained with distilled ImageNet-1K (IPC=10) datasets generated with different merge ratios ($\alpha$), fixed $D_{ds}$=200 and $k$=3. The efficiency is measured in seconds per image generation. **Left** presents the ablation results for single-model distillation, while **Right** shows the corresponding results for multi-model distillation.

| $k$ | 1 | 2 | 3 | 4 | 5 | 6 |
|---|---|---|---|---|---|---|
| ImageNet-1K | 47.1 | 47.6 | **47.8** | 47.6 | 47.4 | 47.3 |
| CIFAR-100 | 59.2 | 60.9 | **61.5** | 60.5 | 59.4 | 57.9 |

| $D_{ds}$ | 160 | 180 | 200 | 224 |
|---|---|---|---|---|
| Post Eval (%) | 47.2 | 47.5 | 47.7 | 47.7 |
| Time Cost (s) | 0.42 | 0.44 | 0.47 | 0.52 |

Table 7: **Left** presents the ablation results for $k$ (frequency of residual connections) using **FADRM** with $D_{ds}$=200, $\alpha$=0.5, while **Right** shows the ablation results for $D_{ds}$ on ImageNet-1K IPC=10. Efficiency is measured as the total time required to optimize a single image under a fixed budget of 2,000 optimization iterations using **FADRM** with $\alpha$=0.5, $k$=3.

**Impact of Components in FADRM.** To assess the contribution of each component (*MRO* and *ARC*), we conduct a comprehensive ablation study. As shown in Table 6, incorporating *MRO* initially leads to a performance drop relative to the baseline (w/o *ARC* and *MRO*). This degradation primarily stems from the loss of critical details during resampling. However, integrating *ARC* and reducing the merge ratio $\alpha$, thereby prioritizing original patches during merging, substantially enhances performance relative to the baseline, which remains susceptible to information vanishing. The optimal performance is observed at a merge ratio of 0.5 for both settings, suggesting that an equal fusion of original and intermediate optimized patches yields the most favorable results. Crucially, the findings clearly show that *ARC* effectively mitigates information vanishing and robustly recovers missing details during resampling, enabling a computationally efficient yet highly effective framework.

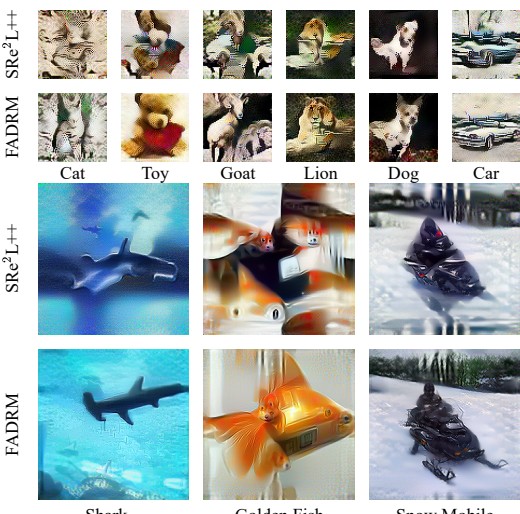

Figure 5: Visualization of dataset distilled by **FADRM** and SRe$^2$L++ on Tiny-ImageNet (top two rows) and ImageNet-1K (bottom two rows).

**Impact of varying $k$.** To examine the effect of $k$ on distilled dataset quality, we perform an ablation study shown in Table 7 (Left). The results indicate that $k = 3$ achieves the best performance. Accuracy improves as $k$ increases from 1 to 3 but drops beyond this point, suggesting that excessive residual connections introduce redundant information and hinder the learning of coherent structures.

**Impact of Downsampled Input Size in *MRO*.** To determine the optimal downsampled input size ($D_{ds}$) for *MRO*, we conduct an ablation study, as presented in Table 7 (Right). Our results demonstrate that $D_{ds}$=200 achieves the most optimal performance. Notably, using other sizes leads to a degradation in the quality of the distilled dataset compared to optimizing with the original input size of 224.

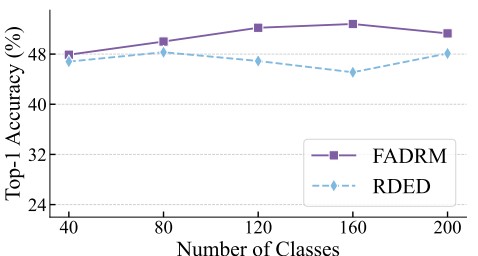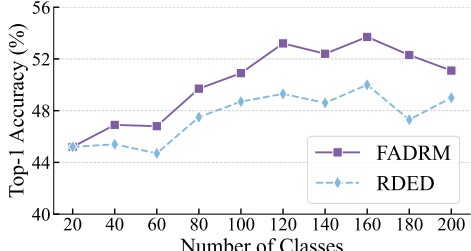

Figure 6: Five-step and Ten-step class-incremental learning on Tiny-ImageNet with IPC=50.

## 4.4 Further Analysis

**Cross-Architecture Generalization.** A fundamental criterion for evaluating the quality of distilled data is its ability to generalize across diverse network architectures, which significantly enhances its practical utility in real-world applications. As illustrated in Table 3 (Right), **FADRM+** consistently outperforms all existing state-of-the-art methods across models of varying sizes and complexities, demonstrating superior generalization capabilities and robustness in diverse scenarios.

**Empirical Rademacher Complexity.** A key assumption (Assumption 10) of our framework is that the empirical Rademacher complexity of the optimized patches exceeds that of the original ones. To verify this, we conduct a detailed empirical analysis. As shown in Table 8, the empirical Rademacher complexity of the optimized patches consistently surpasses that of the originals, thereby supporting our theoretical claim and performance gains.

| Models | $\widehat{\mathfrak{R}}_n(\mathcal{H} \circ \mathcal{O})$ | $\widehat{\mathfrak{R}}_n(\mathcal{H} \circ \tilde{\mathcal{C}}^{res})$ |
|---|---|---|
| ResNet-18 | $4.89 \pm 0.05$ | $4.94 \pm 0.04$ |
| MobileNet-V2 | $4.76 \pm 0.01$ | $4.78 \pm 0.01$ |
| ShuffleNet-V2 | $4.72 \pm 0.02$ | $4.75 \pm 0.01$ |

Table 8: Empirical Rademacher Complexity of optimized vs. original patches across different architectures.

## 4.5 Distilled Image Visualization

Fig. 5 compares distilled data from **FADRM** and SRe$^2$L++ [6], both using ResNet-18 with identical initial patch images, differing only in **FADRM**'s incorporation of residual connections. As shown, **FADRM** more faithfully and effectively preserves the critical features of the original patches and consistently and visibly retains significantly more details than SRe$^2$L++. This highlights the advantage of residual connections in enhancing information density and improving the quality of distilled data.

## 4.6 Application: Continual Learning

Leveraging continual learning to verify the effectiveness of distilled dataset generalization has been widely used in prior work [43, 30, 46]. Following these protocols and utilizing the class-incremental learning framework as in DM [46], we conduct an evaluation on Tiny-ImageNet IPC=50 using a 5-step and 10-step incremental setting, as shown in Fig. 6. The results clearly indicate that **FADRM** consistently surpasses RDED in various settings, demonstrating its effectiveness.

## 5 Conclusion

We proposed **FADRM**, a novel framework for dataset distillation designed to generate high-quality distilled datasets with significantly reduced computational overhead. Our work identifies and addresses the critical challenge of vanishing information, a fundamental limitation in *Uni-Level Framework* that heavily undermines the information density of distilled datasets. To address this, we introduce *data-level residual connections*, a novel mechanism that balances the operations of preserving critical original features and integrating new information, enriching the distilled dataset with both original and new condensed features and increasing its overall information density. Furthermore, by integrating parameter mixed precision training and input multi-resolution optimization, our framework achieves significant reductions in both Peak GPU memory consumption and training time. Extensive experiments demonstrate that **FADRM** outperforms existing state-of-the-art methods in both efficiency and accuracy across multiple benchmark datasets. For future work, we aim to extend the idea of *data-level residual connections* to broader modalities and applications of dataset distillation tasks.

## Acknowledgments

This work was supported by the MBZUAI–WIS Joint Program for AI Research.

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

# Appendix of FADRM

## Contents

# A  Theoretical Derivation

## A.1  Preliminary

**Lemma 1** (Data Processing Inequality [3])**.** *Let $X \to Y \to Z$ form a Markov chain. Then the mutual information between $X$ and $Z$ is upper bounded by that between $X$ and $Y$:*

$$I(X; Z) \leq I(X; Y). \tag{12}$$

*In particular, no post-processing of $Y$ can increase the information that $Y$ contains about $X$.*

**Theorem 4** (Temperature-scaled KL divergence is bounded and Lipschitz-continuous)**.** *Fix integers $k \geq 2$ and a constant $C > 0$. For any temperature $T > 0$ let*

$$z = (z_1, \dots, z_k) \in [-C, C]^k, \qquad q_i^{(T)} = \frac{\exp\!\big(z_i/T\big)}{\displaystyle\sum_{j=1}^{k} \exp\!\big(z_j/T\big)} \quad (i = 1, \dots, k). \tag{13}$$

*Let $p = (p_1, \dots, p_k) \in \Delta_k$ be an arbitrary target probability vector (e.g. it may come from another soft-max with its own temperature). Define the loss*

$$\ell\big(p, q^{(T)}(z)\big) := \mathrm{KL}\big(p \parallel q^{(T)}(z)\big) = \sum_{i=1}^{k} p_i \log \frac{p_i}{q_i^{(T)}}. \tag{14}$$

*Then the following hold:*

1. *(Bounded range) For every admissible pair $(p, z)$,*

$$0 \;\leq\; \ell\big(p, q^{(T)}(z)\big) \;\leq\; B, \qquad B := \log k + \frac{2C}{T}. \tag{15}$$

2. *($\ell_\infty$-Lipschitz continuity in logits) The map $z \mapsto \ell\big(p, q^{(T)}(z)\big)$ is $L$-Lipschitz w.r.t. the $\ell_\infty$ norm with $L = \frac{1}{T}$. Consequently, it is $\sqrt{k}/T$-Lipschitz w.r.t. the Euclidean norm.*

*proof of Theorem 4.* **(i) Boundedness.** Write

$$\mathrm{KL}\big(p \parallel q^{(T)}\big) = \sum_{i=1}^{k} p_i \log p_i - \sum_{i=1}^{k} p_i \log q_i^{(T)}. \tag{16}$$

Since $x \mapsto x \log x$ is non-positive on $[0, 1]$, the first term is at most 0, so

$$\mathrm{KL}\big(p \parallel q^{(T)}\big) \leq - \sum_{i=1}^{k} p_i \log q_i^{(T)}. \tag{17}$$

For the soft-max, $\log q_i^{(T)} = z_i/T - \log Z$, where $Z := \sum_{j=1}^{k} \exp(z_j/T)$. Hence

$$- \sum_{i=1}^{k} p_i \log q_i^{(T)} = -\frac{1}{T} \sum_{i=1}^{k} p_i z_i + \log Z. \tag{18}$$

Because each $z_i \in [-C, C]$ and $\sum_i p_i = 1$,

$$-\frac{1}{T} \sum_i p_i z_i \;\leq\; \frac{C}{T}. \tag{19}$$

Moreover, $z_i \leq C$ implies $Z \leq k \exp(C/T)$ and thus $\log Z \leq \log k + \frac{C}{T}$. Combining the two parts yields the desired upper bound $\log k + 2C/T$. Non-negativity of KL divergence gives the lower bound 0.

**(ii) Lipschitz continuity.** Differentiate $\ell$ w.r.t. $z_i$:

$$\partial_{z_i}\ell\big(p, q^{(T)}(z)\big) = -\frac{p_i - q_i^{(T)}}{T}. \tag{20}$$

Because $|p_i - q_i^{(T)}| \leq 1$, we have $|\partial_{z_i}\ell| \leq 1/T$ for every coordinate. Thus $\|\nabla_z\ell\|_\infty \leq 1/T$, and by the mean-value theorem,

$$\big|\ell(p, q^{(T)}(z)) - \ell(p, q^{(T)}(z'))\big| \leq \frac{1}{T}\|z - z'\|_\infty, \quad \forall z, z' \in [-C, C]^k, \tag{21}$$

so $L = 1/T$ in the $\ell_\infty$ norm. Since $\|v\|_2 \leq \sqrt{k}\|v\|_\infty$, the Euclidean Lipschitz constant is at most $\sqrt{k}/T$. $\qquad\square$

**Lemma 2** (Generalization Bound via Rademacher Complexity [2]). *Let $\mathcal{H}$ be a class of functions mapping $\mathcal{X} \to [0, B]$, and let $S = \{x_1, \ldots, x_n\}$ be an i.i.d. sample from distribution $\mathcal{D}$. Then, for any $\delta > 0$, with probability at least $1 - \delta$, the following inequality holds for all $h \in \mathcal{H}$:*

$$\mathbb{E}_{x\sim\mathcal{D}}[h(x)] \leq \frac{1}{n}\sum_{i=1}^{n} h(x_i) + 2\Re_n(\mathcal{H}) + B\sqrt{\frac{\log(1/\delta)}{2n}} \tag{22}$$

**Lemma 3** (Empirical Risk Proximity). *Let $\tilde{x}_i := \alpha\tilde{x}_i^{\mathrm{res}} + (1 - \alpha)x_i$ with $\alpha \in (0, 1)$, and let the corresponding datasets be $\tilde{\mathcal{C}}^{\mathrm{res}} := \{(\tilde{x}_i^{\mathrm{res}}, y_i)\}_{i=1}^n$, $\tilde{\mathcal{C}}_{\mathrm{FADRM}} := \{(\tilde{x}_i, y_i)\}_{i=1}^n$. Then for any model $h \in \mathcal{H}$, the empirical risk difference is bounded by a negligible value:*

$$\left|\widehat{\mathcal{L}}_{\mathrm{res}}(h) - \widehat{\mathcal{L}}_{\mathrm{FADRM}}(h)\right| \leq L_\ell L_h(1 - \alpha) \cdot \Delta_1, \quad \text{where } \Delta_1 := \frac{1}{n}\sum_{i=1}^{n}\|\tilde{x}_i^{\mathrm{res}} - x_i\|. \tag{23}$$

*Proof of Lemma 3.* We begin by computing the pointwise difference in the loss:

$$|\ell(h(\tilde{x}_i^{\mathrm{res}}), y_i) - \ell(h(\tilde{x}_i), y_i)|. \tag{24}$$

Since $\ell$ is $L_\ell$-Lipschitz in the model output, and $h$ is $L_h$-Lipschitz in the input, we have:

$$|\ell(h(\tilde{x}_i^{\mathrm{res}}), y_i) - \ell(h(\tilde{x}_i), y_i)| \leq L_\ell \cdot |h(\tilde{x}_i^{\mathrm{res}}) - h(\tilde{x}_i)| \leq L_\ell L_h \cdot \|\tilde{x}_i^{\mathrm{res}} - \tilde{x}_i\|. \tag{25}$$

Note that:

$$\tilde{x}_i = \alpha\tilde{x}_i^{\mathrm{res}} + (1 - \alpha)x_i \quad \Rightarrow \quad \tilde{x}_i^{\mathrm{res}} - \tilde{x}_i = (1 - \alpha)(\tilde{x}_i^{\mathrm{res}} - x_i), \tag{26}$$

so:

$$\|\tilde{x}_i^{\mathrm{res}} - \tilde{x}_i\| = (1 - \alpha)\|\tilde{x}_i^{\mathrm{res}} - x_i\|. \tag{27}$$

Therefore,

$$|\ell(h(\tilde{x}_i^{\mathrm{res}}), y_i) - \ell(h(\tilde{x}_i), y_i)| \leq L_\ell L_h(1 - \alpha)\|\tilde{x}_i^{\mathrm{res}} - x_i\|. \tag{28}$$

Averaging over $n$ samples:

$$\left|\widehat{\mathcal{L}}_{\mathrm{res}}(h) - \widehat{\mathcal{L}}_{\mathrm{FADRM}}(h)\right| \leq \frac{1}{n}\sum_{i=1}^{n} L_\ell L_h(1 - \alpha)\|\tilde{x}_i^{\mathrm{res}} - x_i\| = L_\ell L_h(1 - \alpha) \cdot \Delta_1. \tag{29}$$

$$\square$$

**Corollary 1** (Lipschitz Convex Combination Bound). *Let $h : \mathbb{R}^d \to \mathbb{R}$ be an L-Lipschitz function. For any $x, y \in \mathbb{R}^d$ and $\alpha \in (0, 1)$, define $z = \alpha x + (1 - \alpha)y$. Then:*

$$|h(z) - (\alpha h(x) + (1 - \alpha)h(y))| \leq L\alpha(1 - \alpha)\|x - y\| \tag{30}$$

*In particular, this implies:*

$$h(z) \leq \alpha h(x) + (1 - \alpha)h(y) + L\alpha(1 - \alpha)\|x - y\| \tag{31}$$

$$h(z) \geq \alpha h(x) + (1 - \alpha)h(y) - L\alpha(1 - \alpha)\|x - y\| \tag{32}$$

## A.2 Bounded Information in BN-Aligned Synthetic Data

*Proof of Theorem 1.* Let $\mathcal{O}$ denote the original dataset. From it, a pretrained model $f_\theta$ is derived, which includes BatchNorm statistics $\{\mu_l, \sigma_l^2\}$. Each synthetic image $\tilde{x}_j$ in the distilled dataset $\mathcal{C}$ is generated by minimizing an objective function depending only on $f_\theta$ and a fixed label $\tilde{y}_j$.

We assume that each $\tilde{x}_j$ is generated independently given $f_\theta$, and that $f_\theta$ is a deterministic function of $\mathcal{O}$. Then, for each sample $(\tilde{x}_j, \tilde{y}_j)$, we have the Markov chain:

$$\mathcal{O} \to f_\theta \to \tilde{x}_j, \tag{33}$$

By applying Lemma 1, we get:

$$I(\tilde{x}_j; \mathcal{O}) \leq I(f_\theta; \mathcal{O}) = H(f_\theta), \tag{34}$$

Now, by the chain rule of mutual information:

$$I(\mathcal{C}; \mathcal{O}) = I(\{\tilde{x}_j, \tilde{y}_j\}_{j=1}^{|\mathcal{C}|}; \mathcal{O}) \leq \sum_{j=1}^{|\mathcal{C}|} I(\tilde{x}_j; \mathcal{O}) \leq |\mathcal{C}| \cdot H(f_\theta), \tag{35}$$

where we used the fact that $\tilde{y}_j$ is fixed and independent of $\mathcal{O}$ and the independence assumption across samples. Thus, the total information that the synthetic dataset $\mathcal{C}$ can retain about the original dataset $\mathcal{O}$ is bounded by the product of its size and the entropy of the model $f_\theta$. □

## A.3 *ARC* improves the robustness of the distilled images

We evaluate the performance gain of **FADRM** over the uni-level framework from two complementary perspectives: (1) **Generalization Bound Analysis**, and (2) **Data Processing Inequality**.

### A.3.1 Generalization Bound Analysis.

We first analyze the advantage of **FADRM** through the lens of generalization bounds. The central insight of this analysis is that when the synthetic data is optimized, exhibiting low cross-entropy loss and strong feature alignment, it becomes overly easy for the model to fit. Such data provides limited regularization, making the student model susceptible to overfitting and resulting in degraded generalization to unseen samples. In contrast, when synthetic data is combined with real data under the **FADRM** framework, the real data introduces natural variability and subtle distributional nuances that act as an implicit regularizer. This hybrid training scheme mitigates overfitting by constraining the hypothesis space, thereby improving the model's generalization performance.

**Analysis of Assumption 10.** To facilitate subsequent theoretical analysis, we first provide a preliminary interpretation of Assumption 10. We begin with a mild and empirically supported assumption that the Rademacher complexity of the optimized patches exceeds that of the original ones, i.e.,

$$\mathfrak{R}_n(\mathcal{H} \circ \tilde{\mathcal{C}}^{\text{res}}) > \mathfrak{R}_n(\mathcal{H} \circ \mathcal{O}), \tag{36}$$

where $\mathcal{O}$ denotes the original data patch and $\tilde{\mathcal{C}}^{\text{res}}$ represents its optimized counterpart. Building upon this preliminary inequality, we introduce small constants to achieve a more general analytical form, taking into account the bounded smoothness and Lipschitz continuity of $\ell$ and $h$. In practice, these constants remain small due to the use of high-temperature KL loss and strong regularization (e.g., CutMix). Accordingly, the assumption can be equivalently expressed as

$$\mathfrak{R}_n(\mathcal{H} \circ \tilde{\mathcal{C}}^{\text{res}}) - \mathfrak{R}_n(\mathcal{H} \circ \mathcal{O}) > \left(\frac{L_h L_\ell}{2B} + \frac{L_h}{2}\right) \cdot \Delta, \tag{37}$$

where $L_\ell$ and $L_h$ denote the Lipschitz constants of the loss and hypothesis functions, respectively. Since both $L_h$ and $L_\ell$ are small in typical settings, the coefficient $\left(\frac{L_h L_\ell}{2B} + \frac{L_h}{2}\right)$ becomes negligible, allowing us to simplify subsequent theoretical derivations.

*Proof of Theorem 2.* Let $\tilde{x}_i^{res}$ be a perturbation generated via distribution (running statistics) matching and prediction (cross entropy) matching, and let $x_i$ be a real image from the original dataset.

Define the residual-injected sample $\tilde{x}_i$ as:

$$\tilde{x}_i := \alpha \tilde{x}_i^{res} + (1 - \alpha)x_i, \quad \alpha \in (0, 1) \tag{38}$$

Define the datasets:

- $\tilde{\mathcal{C}}^{res} = \{\tilde{x}_i^{res}, \tilde{y}_i^{res}\}_{i=1}^n$: perturbation generated via distribution (running statistics) matching and prediction (cross entropy) matching,

- $\mathcal{O} = \{x_i, y_i\}_{i=1}^n$: selected patches from the original dataset,

- $\tilde{\mathcal{C}}_{\text{FADRM}} = \{\tilde{x}_i, \tilde{y}_i\}_{i=1}^n$: residual-injected dataset.

We begin by bounding the Rademacher complexity of the residual-injected dataset $\tilde{\mathcal{C}}_{\text{FADRM}} = \{\tilde{x}_i\}_{i=1}^n$, where $\tilde{x}_i = \alpha\tilde{x}_i^{res} + (1-\alpha)x_i$, using Lemma 2 and Corollary 1.

From the definition:

$$\mathfrak{R}_n(\mathcal{H} \circ \tilde{\mathcal{C}}_{\text{FADRM}}) = \mathbb{E}_{\boldsymbol{\sigma}}\left[\sup_{h \in \mathcal{H}} \frac{1}{n}\sum_{i=1}^n \sigma_i h(\tilde{x}_i)\right] \tag{39}$$

By Corollary 1, we have for each term:

$$h(\tilde{x}_i) \leq \alpha h(\tilde{x}_i^{res}) + (1-\alpha)h(x_i) + \varepsilon_i, \quad \text{where } |\varepsilon_i| \leq L_h \cdot \alpha(1-\alpha)\|\tilde{x}_i^{res} - x_i\| \tag{40}$$

Therefore:

$$\sum_{i=1}^n \sigma_i h(\tilde{x}_i) \leq \sum_{i=1}^n \sigma_i\left(\alpha h(\tilde{x}_i^{res}) + (1-\alpha)h(x_i)\right) + \sum_{i=1}^n |\sigma_i\varepsilon_i| \tag{41}$$

Using $|\sigma_i| = 1$, we get:

$$\sum_{i=1}^n |\sigma_i\varepsilon_i| \leq L_h\alpha(1-\alpha)\sum_{i=1}^n \|\tilde{x}_i^{res} - x_i\| = n \cdot L_h\alpha(1-\alpha) \cdot \Delta \tag{42}$$

Divide by $n$, take supremum and expectation:

$$\mathfrak{R}_n(\mathcal{H} \circ \tilde{\mathcal{C}}_{\text{FADRM}}) \leq \alpha \cdot \mathfrak{R}_n(\mathcal{H} \circ \tilde{\mathcal{C}}^{res}) + (1-\alpha) \cdot \mathfrak{R}_n(\mathcal{H} \circ \mathcal{O}) + L_h\alpha(1-\alpha) \cdot \Delta \tag{43}$$

Rearrange the Inequality:

$$\mathfrak{R}_n(\mathcal{H} \circ \tilde{\mathcal{C}}_{\text{FADRM}}) - \mathfrak{R}_n(\mathcal{H} \circ \tilde{\mathcal{C}}^{\text{res}}) \leq (1-\alpha)\left[\mathfrak{R}_n(\mathcal{H} \circ \mathcal{O}) - \mathfrak{R}_n(\mathcal{H} \circ \tilde{\mathcal{C}}^{\text{res}})\right] + L_h\alpha(1-\alpha) \cdot \Delta \tag{44}$$

Multiply $2B$ on both sides and add a negligible positive value $\epsilon$ to the LHS:

$$2B \cdot \left[\mathfrak{R}_n(\mathcal{H} \circ \tilde{\mathcal{C}}_{\text{FADRM}}) - \mathfrak{R}_n(\mathcal{H} \circ \tilde{\mathcal{C}}^{\text{res}})\right] < 2B(1-\alpha) \cdot \left[\mathfrak{R}_n(\mathcal{H} \circ \mathcal{O}) - \mathfrak{R}_n(\mathcal{H} \circ \tilde{\mathcal{C}}^{\text{res}})\right]$$
$$+ 2BL_h\alpha(1-\alpha) \cdot \Delta + \epsilon \tag{45}$$

As validated in Theorem 4, when $T > 0$, KL-divergence becomes a bounded $B$-range loss, which we then apply Lemma 2 to formulate generalization error:

$$\mathcal{L}_{\text{gen}}(h) \leq \widehat{\mathcal{L}}(h) + 2B \cdot \mathfrak{R}_n(\mathcal{H} \circ S) \tag{46}$$

Apply to both models:

$$\mathcal{L}_{\text{gen}}(h_{\text{res}}) \leq \widehat{\mathcal{L}}_{\text{res}} + 2B \cdot \mathfrak{R}_n(\mathcal{H} \circ \tilde{\mathcal{C}}^{res}) \tag{47}$$

$$\mathcal{L}_{\text{gen}}(h_{\text{FADRM}}) \leq \widehat{\mathcal{L}}_{\text{FADRM}} + 2B \cdot \mathfrak{R}_n(\mathcal{H} \circ \tilde{\mathcal{C}}_{\text{FADRM}}) \tag{48}$$

Recall the lower bound for the difference of two ERMs established in Lemma 3, we then have:

$$\widehat{\mathcal{L}}_{\text{res}}(h) - \widehat{\mathcal{L}}_{\text{FADRM}}(h) \geq -L_\ell L_h(1-\alpha) \cdot \Delta, \quad \text{where } \Delta := \frac{1}{n}\sum_{i=1}^n \|\tilde{x}_i^{\text{res}} - x_i\|. \tag{49}$$

Given the Equation (37), we can then derive:

$$-L_\ell L_h(1-\alpha) \cdot \Delta > 2B\left\{(1-\alpha)\left[\mathfrak{R}_n(\mathcal{H} \circ \mathcal{O}) - \mathfrak{R}_n(\mathcal{H} \circ \tilde{\mathcal{C}}^{\text{res}})\right] + L + h\alpha(1-\alpha) \cdot \Delta\right\} + \epsilon \tag{50}$$

where the RHS in Equation (50) is the upper bound for the difference in Rademacher Complexity, we then derive the following inequality:

$$\widehat{\mathcal{L}}_{\text{res}} - \widehat{\mathcal{L}}_{\text{FADRM}} > 2B \cdot \left[ \mathfrak{R}_n(\mathcal{H} \circ \tilde{\mathcal{C}}_{\text{FADRM}}) - \mathfrak{R}_n(\mathcal{H} \circ \tilde{\mathcal{C}}^{res}) \right] \tag{51}$$

which shows:

$$\widehat{\mathcal{L}}_{\text{res}} + 2B \cdot \mathfrak{R}_n(\mathcal{H} \circ \tilde{\mathcal{C}}^{res}) > \widehat{\mathcal{L}}_{\text{FADRM}} + 2B \cdot \mathfrak{R}_n(\mathcal{H} \circ \tilde{\mathcal{C}}_{\text{FADRM}}) \tag{52}$$

$\square$

**Analysis of Merge Ratio $\alpha$.** We further conduct a formal analysis to determine the optimal merge ratio $\alpha$. As demonstrated in the following analysis, the optimal configuration is achieved when $\alpha$ approaches $0.5$, yielding a balanced contribution between the two distillation components and promoting both stable convergence and improved generalization. To theoretically substantiate this choice, we examine the influence of $\alpha$ on the generalization bound of FADRM, defined as follows:

$$D(\alpha) = \widehat{\mathcal{L}}_{\text{FADRM}} + 2B \cdot \mathfrak{R}_n(\mathcal{H} \circ \tilde{\mathcal{C}}_{\text{FADRM}}), \tag{53}$$

where $\widehat{\mathcal{L}}_{\text{FADRM}}$ denotes the empirical loss, and $\mathfrak{R}_n(\cdot)$ is the empirical Rademacher complexity.

Assuming the loss function is $\lambda$-strongly convex and denoting $\Delta := \frac{1}{n} \sum_i \|\tilde{x}_i^{\text{res}} - x_i\|$, we obtain:

$$\widehat{\mathcal{L}}_{\text{FADRM}} \leq \alpha \widehat{\mathcal{L}}_{\text{res}} + (1 - \alpha)\widehat{\mathcal{L}}_{\mathcal{O}} - \frac{\lambda}{2}\alpha(1 - \alpha)\Delta, \tag{54}$$

where $\widehat{\mathcal{L}}_{\text{res}}$ and $\widehat{\mathcal{L}}_{\mathcal{O}}$ are the empirical losses of the residual and original components, respectively.

Following the derivation in Equation 43, the empirical Rademacher complexity can be bounded as:

$$\mathfrak{R}_n(\mathcal{H} \circ \tilde{\mathcal{C}}_{\text{FADRM}}) \leq \alpha \mathfrak{R}_n(\mathcal{H} \circ \tilde{\mathcal{C}}_{\text{res}}) + (1 - \alpha)\mathfrak{R}_n(\mathcal{H} \circ \mathcal{O}) + L_h \cdot \alpha(1 - \alpha)\Delta, \tag{55}$$

where $L_h$ is the Lipschitz constant of the hypothesis class $\mathcal{H}$. Substituting both inequalities, we obtain an upper bound on $D(\alpha)$:

$$D(\alpha) \leq X\alpha + Y(1 - \alpha) + Z(\alpha - 1)\alpha, \tag{56}$$

where the coefficients are defined as:

$$X := \widehat{\mathcal{L}}_{\text{res}} + 2B \cdot \mathfrak{R}_n(\mathcal{H} \circ \tilde{\mathcal{C}}_{\text{res}}),$$
$$Y := \widehat{\mathcal{L}}_{\mathcal{O}} + 2B \cdot \mathfrak{R}_n(\mathcal{H} \circ \mathcal{O}),$$
$$Z := \frac{\lambda}{2}\Delta - 2BL_h\Delta.$$

Minimizing the quadratic bound $D(\alpha)$ with respect to $\alpha$ yields:

$$\alpha^\star = \frac{1}{2} + \frac{Y - X}{2Z}. \tag{57}$$

In practice, FADRM employs a temperature-scaled KL divergence, under which the strong convexity constant satisfies

$$\lambda \approx 1 + (C - 1)\exp\left(-\frac{d}{T}\right), \tag{58}$$

where $C$ is the number of classes, $d$ the embedding dimensionality, and $T$ the temperature parameter. As $T$ increases, $\lambda$ grows approximately linearly with $C$, and the bias term $\frac{Y-X}{2Z}$ becomes negligible compared to the dominant $\lambda$ term. Therefore, the optimal merge ratio $\alpha^\star$ approaches $0.5$, implying that equal weighting between the residual (synthetic) and original (real) components achieves near-optimal generalization. This theoretical conclusion aligns with our empirical findings in Table 6, where $\alpha = 0.5$ consistently yields strong performance across datasets.

### A.3.2 Data Processing Inequality Analysis

We further show that **FADRM** will improve the bound established in Theorem 1.

*Proof of Theorem 3.* We analyze the effect of **FADRM**, particularly the Ajustable Residual Connection (*ARC*) component, in mitigating the limitations of the uni-level framework.

In **FADRM**, the generated image $\hat{x}$ is produced using both the learned model $f_\theta$ and real image patches from $\mathcal{O}$, by iteratively fusing these patches into the intermediate optimized image. This process introduces a new dependency structure:

$$\mathcal{O} \rightarrow \hat{x} \leftarrow f_\theta. \tag{59}$$

As a result, the conditional independence assumption

$$\hat{x} \perp\!\!\!\perp \mathcal{O} \mid f_\theta \tag{60}$$

no longer holds. Consequently, the Markov chain underlying the standard Data Processing Inequality (DPI) for the uni-level framework is violated.

By the chain rule of mutual information, we have:

$$I(\hat{x}; \mathcal{O}) = I(f_\theta; \hat{x}) + I(\hat{x}; \mathcal{O} \mid f_\theta). \tag{61}$$

In the uni-level framework, the generation process depends solely on $f_\theta$, implying

$$I(\hat{x}; \mathcal{O} \mid f_\theta) = 0. \tag{62}$$

However, in the FADRM setting, real image patches directly influence the generation of $\hat{x}$, leading to a strictly positive conditional mutual information term:

$$I(\hat{x}; \mathcal{O} \mid f_\theta) > 0. \tag{63}$$

Therefore,

$$I(\hat{x}_{\mathrm{fadrm}}; \mathcal{O}) = I(f_\theta; \hat{x}) + I(\hat{x}; \mathcal{O} \mid f_\theta) > I(f_\theta; \hat{x}) = I(\hat{x}_{\mathrm{uni}}; \mathcal{O}), \tag{64}$$

which proves the theorem. $\square$

This result demonstrates that FADRM fundamentally breaks the conditional independence constraint inherent in the uni-level framework, allowing additional information from the original dataset $\mathcal{O}$ to flow into the generated samples. Consequently, FADRM captures richer and more diverse information, leading to superior knowledge distillation performance.

## B  Empirical Rademacher Complexity Estimation

**Formulation.** Given a fixed dataset $\mathcal{O} = \{x_i\}_{i=1}^n$ and a hypothesis class $\mathcal{H} = \{h_\theta : \mathcal{X} \rightarrow \mathbb{R}\}$, the empirical Rademacher complexity quantifies the expressive capacity of $\mathcal{H}$ on $\mathcal{O}$:

$$\mathfrak{R}_n(\mathcal{H} \circ \mathcal{O}) = \mathbb{E}_{\boldsymbol{\sigma}} \left[ \sup_{h \in \mathcal{H}} \frac{1}{n} \sum_{i=1}^n \sigma_i h(x_i) \right], \qquad \sigma_i \overset{\text{i.i.d.}}{\sim} \mathrm{Unif}\{-1, +1\}. \tag{65}$$

Here, $\boldsymbol{\sigma} = (\sigma_1, \ldots, \sigma_n)$ is a vector of independent Rademacher random variables representing a random $\pm 1$ labeling of the data. From a data-centric perspective, for a fixed hypothesis class $\mathcal{H}$, $\mathfrak{R}_n(\mathcal{H} \circ \mathcal{O})$ measures how easily the data $\mathcal{O}$ admits spurious alignment with random $\pm 1$ labels; a larger value indicates that the same model family can fit these data more readily, implying higher overfitting propensity and weaker implicit regularization of the dataset.

**Functional interpretation.** For a fixed random draw $\boldsymbol{\sigma}$, define the functional

$$\phi(\boldsymbol{\sigma}) = \sup_{h \in \mathcal{H}} \frac{1}{n} \sum_{i=1}^n \sigma_i h(x_i), \tag{66}$$

which captures the *maximum correlation* between the random sign pattern $\boldsymbol{\sigma}$ and the predictions $h(x_i)$ over all admissible $h \in \mathcal{H}$. In other words, $\phi(\boldsymbol{\sigma})$ represents the single-trial response of $\mathcal{H}$ to a random noise labeling of $\mathcal{O}$. A function class with high capacity will yield a larger $\phi(\boldsymbol{\sigma})$ since it can align more closely with arbitrary noise patterns.

**Monte Carlo approximation.** Because the expectation in Eq. (66) is intractable, we estimate it via $M$ i.i.d. Monte Carlo samples $\{\boldsymbol{\sigma}^{(m)}\}_{m=1}^{M}$:

$$\widehat{\mathfrak{R}}_n^{(M)}(\mathcal{H}\circ\mathcal{O}) \ = \ \frac{1}{M}\sum_{m=1}^{M}\phi(\boldsymbol{\sigma}^{(m)}), \qquad \phi(\boldsymbol{\sigma}^{(m)}) \ = \ \sup_{h\in\mathcal{H}}\frac{1}{n}\sum_{i=1}^{n}\sigma_i^{(m)}h(x_i). \tag{67}$$

Each $\phi(\boldsymbol{\sigma}^{(m)})$ can be viewed as one stochastic probe of the capacity of $\mathcal{H}$, and their empirical average approximates the true expectation $\mathbb{E}_{\boldsymbol{\sigma}}[\phi(\boldsymbol{\sigma})]$.

**Optimization-based inner supremum.** The inner supremum $\sup_{h\in\mathcal{H}}$ in Eq. (66) is approximated via empirical optimization. Given a parameterized model $h_\theta(x) = B\tanh(g_\theta(x))$ bounded to $[-B, B]$, we maximize $\Phi(\theta;\boldsymbol{\sigma}) = \frac{1}{n}\sum_i \sigma_i h_\theta(x_i)$ by minimizing its negation:

$$\mathcal{L}(\theta;\boldsymbol{\sigma}) \ = \ -\frac{1}{n}\sum_{i=1}^{n}\sigma_i h_\theta(x_i). \tag{68}$$

We train $\theta$ for $E$ epochs using SGD or Adam and record the best achieved value

$$\widehat{\phi}^\star(\boldsymbol{\sigma}) \ = \ \max_{1\le e\le E}\frac{1}{n}\sum_{i=1}^{n}\sigma_i\, h_{\theta^{(e)}}(x_i) \ \approx \ \sup_\theta\frac{1}{n}\sum_{i=1}^{n}\sigma_i h_\theta(x_i). \tag{69}$$

Since the optimization may not reach the global maximum, $\widehat{\phi}^\star(\boldsymbol{\sigma})$ serves as a *lower bound* on the true supremum, and the final $\widehat{\mathfrak{R}}_n$ estimate is thus a conservative (under-)approximation of the true complexity.

**Empirical estimation for optimized and original patches.** For the optimized image patches $\tilde{\mathcal{C}}^{\text{res}}$, we compute an analogous quantity:

$$\widehat{\mathfrak{R}}_n(\mathcal{H}\circ\tilde{\mathcal{C}}^{\text{res}}) \ = \ \frac{1}{M}\sum_{m=1}^{M}\max_{1\le e\le E}\frac{1}{n}\sum_{i=1}^{n}\sigma_i^{(m)}\, h_{\theta^{(e)}}(\tilde{x}_i^{\text{res}}), \tag{70}$$

and compare it to that computed on the original patches $\mathcal{O}$. Under the theoretical assumption in Eq. 10, if the residual optimization step enriches the feature expressivity of the synthetic data, we expect the following relationship to hold:

$$\widehat{\mathfrak{R}}_n(\mathcal{H}\circ\tilde{\mathcal{C}}^{\text{res}}) \ > \ \widehat{\mathfrak{R}}_n(\mathcal{H}\circ\mathcal{O}). \tag{71}$$

**Statistical reporting.** Let $v_m = \widehat{\phi}^\star(\boldsymbol{\sigma}^{(m)})$ denote the best objective obtained for each Rademacher draw. We report the sample mean and standard error (SE) across $M$ trials:

$$\overline{\mathfrak{R}_n} \ = \ \frac{1}{M}\sum_{m=1}^{M}v_m, \qquad \text{SE} \ = \ \frac{\sqrt{\frac{1}{M-1}\sum_{m=1}^{M}(v_m - \overline{v})^2}}{\sqrt{M-1}}. \tag{72}$$

This Monte Carlo–based estimator provides a reproducible and statistically sound measure of empirical $\mathfrak{R}_n$, thereby linking theoretical generalization guarantees with observed improvements in data-level residual optimization.

## C  Additional Experiments

### C.1  Adaptive Multi-Resolution Optimization

To avoid manually selecting the downsampled dimension $D_{\text{ds}}$, we apply an **adaptive multi-resolution strategy** that automatically adjusts the effective image resolution according to the smoothness of the input. The underlying idea is that smoother images can tolerate stronger downsampling without significant loss of fidelity, whereas images with rich textures require higher resolutions.

We estimate image smoothness using the total variation (TV) of a batch $x \in \mathbb{R}^{B\times C\times H\times W}$:

$$\text{TV}(x) = \frac{1}{B}\sum_{i=1}^{B}\left(\|x_i[:, 1:, :] - x_i[:, :-1, :]\|_1 + \|x_i[:, :, 1:] - x_i[:, :, :-1]\|_1\right). \tag{73}$$

Based on the TV value, we compute an adaptive shrink ratio:

$$\text{ShrinkRatio} = \min\left(0.8, \ \frac{0.5}{\max(\textbf{TV}, 10^{-4})}\right), \tag{74}$$

and the corresponding downsampled dimension:

$$D_{\text{ds}} = \max\left(\lfloor D \cdot \text{ShrinkRatio} \rfloor, \ 128\right). \tag{75}$$

This adaptive adjustment enables automatic control of the resolution level during training without manual tuning. As shown in Table 9, the adaptive strategy achieves comparable performance to manually selected resolutions.

| Dataset | Manually Selected | Adaptive |
|---|---|---|
| ImageNet-1K | 47.7% | 47.5% |

Table 9: Comparison between manually selected and adaptive multi-resolution optimization.

This adaptive method effectively removes the need for manual hyperparameter search while maintaining model performance.

## C.2 Distillation Performance vs. Model Confidence

As shown in Theorem 1, **FADRM**, similar to other uni-level distillation methods, is most effective when the pretrained model is not overly confident, i.e., when its output distribution exhibits higher entropy. In this regime, the mutual information upper bound becomes looser, allowing the distilled data to better capture information from the original dataset.

To empirically verify this relationship, we vary the number of training epochs used to obtain the teacher model, which directly affects its output confidence. As the teacher becomes more confident (lower cross-entropy), the effectiveness of FADRM decreases, consistent with the theoretical prediction. The results are summarized in Table 10.

| Squeezed Epochs | Cross-Entropy | FADRM Accuracy (%) |
|---|---|---|
| 50 | 0.0338 | 61.5 |
| 100 | 0.0054 | 59.6 |
| 150 | 0.0031 | 58.7 |
| 200 | 0.0028 | 58.3 |

Table 10: Relationship between teacher confidence (measured by cross-entropy) and FADRM performance. Lower cross-entropy indicates higher confidence.

These results align with the theoretical insight that moderate uncertainty in the teacher model facilitates more informative knowledge transfer, while overly confident teachers limit the representational diversity distilled into the student.

## C.3 More Experimental Results of ARC on Small Datasets

To further evaluate the performance of ARC on small-scale datasets, we conduct experiments on CIFAR-100. As shown in Table 11, the results indicate that $\alpha = 0.5$ yields the best performance, consistent with our observation on the large-scale dataset (ImageNet-1K).

| $\alpha$ | 0.9 | 0.8 | 0.7 | 0.6 | 0.5 | 0.4 | 0.3 |
|---|---|---|---|---|---|---|---|
| **FADRM Accuracy (%)** | 58.0 | 58.9 | 58.7 | 59.6 | **61.5** | 59.2 | 58.6 |

Table 11: Performance of ARC on CIFAR-100 across different values of $\alpha$. The optimal setting $\alpha = 0.5$ is consistent with the observation on ImageNet-1K.

## C.4 Stability of *ARC*

To verify that ARC maintains semantic consistency rather than introducing disruptive information, we conduct two complementary analyses. First, we compute the cosine similarity between features before and after ARC injection across stages. Second, we assess the accuracy of the distilled images before and after applying ARC using a pretrained model as a verifier. As shown in the Table 12, the feature-level cosine similarity remains high ($\geq 0.88$) across all stages, indicating strong semantic preservation. Although verifier accuracy shows a slight decline (from 100% to 97%) at the initial injection stage, it quickly recovers in subsequent stages, suggesting that ARC adds semantically aligned residual content rather than harmful noise. Overall, these results confirm that incorporating ARC is stable and does not negatively impact the optimization process.

| Stages | Feature Cosine Similarity | Before ARC Accuracy (%) | After ARC Accuracy (%) |
|---|---|---|---|
| Residual Injection Stage #1 | 0.89 | 100 | 97 |
| Residual Injection Stage #2 | 0.88 | 100 | 100 |
| Residual Injection Stage #3 | 0.88 | 100 | 100 |

Table 12: Feature stability and accuracy before and after ARC injection across different residual stages on CIFAR-100.

## C.5 Contribution of Individual Components on Efficiency Gains

To clarify the individual contributions of Mixed Precision Training (MPT) and Multi-Resolution Optimization (MRO) to computational efficiency, we conducted an ablation study as shown in the table below, the experiment is conducted on ImageNet-1K, where efficiency is measured based on the time needed for generating a single distilled image. As shown in Table 13, MRO is primarily intended to reduce the overall wall-clock time required to optimize a single image by iteratively performing training on downsampled versions. Since the process still involves full-resolution updates, the peak GPU usage remains unchanged. In contrast, MPT is designed to reduce both computation time and peak memory consumption by leveraging lower-precision operations throughout.

While MPT accounts for the majority of the efficiency improvement, MRO provides complementary gains. For instance, when comparing the configurations with and without MRO under MPT, we observe a reduction of 0.05 seconds per image. Although this difference may appear marginal at the individual level, it becomes meaningful when scaled. For example, generating a 50 IPC ImageNet-1K dataset (i.e., 50,000 images) results in,

$$50,000 \times 0.05\text{seconds} = 2,500\text{seconds} \approx 42\text{minutes}, \tag{76}$$

In summary, MPT is the primary contributor to efficiency, and MRO further improves performance by iteratively optimizing the images at a smaller resolution.

| Settings | Time Cost (s / img) | Minimum GPU Usage (GB) | Peak GPU Usage (GB) |
|---|---|---|---|
| w/o MPT, w/o MRO | 1.26 | 5.3 | 5.3 |
| w/o MPT, w/ MRO | 0.84 | 4.0 | 5.3 |
| w/ MPT, w/o MRO | 0.52 | 2.9 | 2.9 |
| w/ MPT, w/ MRO | 0.47 | 2.5 | 2.9 |

Table 13: **Efficiency ablation on ImageNet-1K.** Decomposition of the contributions from Mixed Precision Training (MPT) and Multi-Resolution Optimization (MRO). We report wall-clock time per distilled image (s/img) and GPU memory usage. MPT chiefly reduces both time and peak memory, while MRO provides additional wall-clock savings with unchanged peak memory.

## D   Optimization Details

Formally, the optimization process adheres to the principle of aligning the synthesized data with both the predictive behavior and the statistical distribution captured by a pretrained model $f_\theta$. Specifically, given a synthesized image $\tilde{x}_t$ at iteration $t$, the optimization objective is defined as:

$$\underset{\tilde{x}_t}{\arg\min} \quad \mathcal{L}(f_\theta(\tilde{x}_t), \tilde{\mathbf{y}}) + \mathcal{D}_{\text{global}}(\tilde{x}_t), \tag{77}$$

where $l(f_\theta(\tilde{x}_t), \tilde{\mathbf{y}})$ enforces consistency with the target predictions, while $\mathcal{D}_{\text{global}}(\tilde{x}_t)$ ensures alignment with the statistical distribution. Importantly, the parameters of $f_\theta$ remain fixed throughout the optimization, and only $\tilde{x}_t$ is updated.

The prediction alignment term is formulated as the cross-entropy loss computed over the synthesized batch:

$$\mathcal{L}(f_\theta(\tilde{x}_t), \tilde{\mathbf{y}}) = -\frac{1}{N} \sum_{n=1}^{N} \sum_{i=1}^{C} \tilde{\mathbf{y}}_{n,i} \log f_\theta(\tilde{x}_t)_{n,i}, \tag{78}$$

where $N$ denotes the batch size, and $C$ represents the total number of classes. The alignment to the distribution in pretrained model is calculated as follows:

$$\mathcal{D}_{\text{global}}(\tilde{x}_t) = \sum_l \left\| \mu_l(\tilde{x}_t) - \mathbb{E}[\mu_l | \mathcal{O}] \right\|_2$$
$$+ \sum_l \left\| \sigma_l^2(\tilde{x}_t) - \mathbb{E}[\sigma_l^2 | \mathcal{O}] \right\|_2$$
$$= \sum_l \left\| \mu_l(\tilde{x}_t) - \mathbf{BN}_l^{\text{RM}} \right\|_2$$
$$+ \sum_l \left\| \sigma_l^2(\tilde{x}_t) - \mathbf{BN}_l^{\text{RV}} \right\|_2,$$

where $\mathcal{O}$ denotes the original dataset, and $l$ indexes the layers of the model. The terms $\mathbf{BN}_l^{\text{RM}}$ and $\mathbf{BN}_l^{\text{RV}}$ correspond to the running mean and running variance of the Batch Normalization (BN) statistics at layer $l$. By minimizing $\mathcal{D}_{\text{global}}(\tilde{x}_t)$, the synthesized data is encouraged to exhibit statistical characteristics consistent with the original dataset, thereby preserving global information.

## E   Resampling via Bilinear Interpolation

Given an original image $I : \mathbb{Z}^2 \to \mathbb{R}^C$ defined on discrete pixel coordinates, the continuous extension $\tilde{I} : \mathbb{R}^2 \to \mathbb{R}^C$ at non-integer location $(i', j') \in \mathbb{R}^2$ is computed via bilinear interpolation as follows:

$$\tilde{I}(i', j') = \sum_{m=0}^{1} \sum_{n=0}^{1} w_{m,n} \cdot I(i + m, j + n), \tag{79}$$

where $i = \lfloor i' \rfloor$, $j = \lfloor j' \rfloor$, $\alpha = i' - i \in [0, 1)$, $\beta = j' - j \in [0, 1)$, and the interpolation weights are defined by:

$$w_{m,n} = (1 - m + (-1)^m \alpha)(1 - n + (-1)^n \beta). \tag{80}$$

Explicitly, Equation (79) expands to:

$$\tilde{I}(i', j') = (1 - \alpha)(1 - \beta) \cdot I(i, j) + \alpha(1 - \beta) \cdot I(i + 1, j)$$
$$+ (1 - \alpha)\beta \cdot I(i, j + 1) + \alpha\beta \cdot I(i + 1, j + 1), \tag{81}$$

This interpolation scheme can be viewed as a separable approximation to the continuous image function, with weights derived from tensor-product linear basis functions over the unit square. It preserves differentiability with respect to the fractional coordinates $(i', j')$, making it particularly amenable to gradient-based optimization frameworks.

## F   Limitations

While FADRM offers substantial improvements in computational efficiency and performance for dataset distillation, it also introduces several limitations. First, the method relies on the assumption that residual signals between synthetic and real data capture critical learning dynamics, which may not generalize across domains with highly abstract or non-visual modalities such as natural language or time-series data. Second, the use of distilled datasets can inadvertently reinforce biases present in the original data if not carefully audited, potentially leading to fairness concerns in downstream applications. From a broader societal perspective, while FADRM reduces the computation and

resource demands of training large models, thereby contributing positively to sustainability, it may also facilitate the deployment of powerful models in low-resource or surveillance scenarios without adequate ethical oversight. Thus, responsible deployment and continued research into bias mitigation and cross-domain generalization are essential to ensure the safe and equitable application of FADRM.

# G Experimental Setup

## G.1 Experiment Pipeline

Our experiments start by generating synthetic images from pretrained teacher models using the proposed distillation framework. The resulting images form the distilled dataset, which is subsequently used to train a student model from scratch under the training configurations detailed in Appendix G.2. To evaluate the effectiveness of the distilled datasets, we adopt a standard image classification task. Each distilled dataset is used to train a student model from scratch, and the resulting model is evaluated on the original test set of the corresponding dataset (e.g., CIFAR-100, Tiny-ImageNet, ImageNet-1K). The evaluation metric is top-1 classification accuracy, computed using the final checkpoint after full training. This protocol ensures that the measured performance differences stem from the representational fidelity and diversity of the distilled data rather than differences in optimization configurations or model initialization.

## G.2 Hyper-parameters Setting

Our method strictly follows the training configuration established in EDC to ensure a fair and consistent comparison across all evaluated approaches. Additionally, we re-run RDED and CV-DD under the same configuration and report the highest performance obtained between their original setup and the EDC configuration. This methodology guarantees a rigorous and equitable evaluation by accounting for potential variations in training dynamics across different settings.

**Full Dataset Training.** To establish an upper bound on performance across different backbone architectures (representing the results achieved when training models on the full original dataset) we adopt the hyperparameters specified in Table 14. These hyperparameters are carefully chosen to ensure full model convergence while effectively mitigating the risk of overfitting, thereby providing a reliable reference for evaluating the performance of distilled datasets.

| Hyperparameters for Training the Original Dataset | |
|---|---|
| Optimizer | SGD |
| Learning Rate | 0.1 |
| Weight Decay | 1e-4 |
| Momentum | 0.9 |
| Batch Size | 128 |
| Loss Function | Cross-Entropy |
| Epochs | 300 |
| Augmentation | RandomResizedCrop, Horizontal Flip, CutMix |

Table 14: Hyperparameters for Training the Original Dataset.

In summary, the synthesis of distilled data follows consistent hyperparameter configurations, as outlined in Table 15. Variations in hyperparameters are introduced exclusively during two phases: (1) the model Pre-training phase. and (2) the post-evaluation phase. These adjustments are carefully tailored based on the scale of the models and the specific characteristics of the datasets used. During the post-evaluation phase, we have four hyperparameter combinations, as detailed in Table 16. Among these parameters, $\eta$ plays a critical role in controlling the decay rate of the learning rate, as defined by the cosine learning rate schedule in Equation 82. Specifically, a larger value of $\eta$ results in a slower decay rate, thereby preserving a higher learning rate for a longer duration during training.

$$Learning\ Rate = 0.5 \times \left( 1 + \cos \left( \pi \frac{step}{epochs \times \eta} \right) \right) \tag{82}$$

| Hyperparameter | Value |
|---|---|
| Optimizer | Adam |
| Learning rate | 0.25 |
| Beta | (0.5, 0.9) |
| Epsilon | $1 \times 10^{-8}$ |
| Batch Size | 100 or 10 (if $C < 100$) |
| Iterations ($N_{\text{iter}}$) | 2,000 |
| Merge Ratio ($\alpha$) | 0.5 |
| Number of *ARC* ($k$) | 3 |
| Downsampled Size ($D_{\text{ds}}$) | 200 (ImageNet-1K and Its subsets), Original Input Size (CIFAR-100, Tiny-ImageNet) |
| **FADRM** Recover Model | ResNet18 |
| **FADRM+** Recover Model | ResNet18 DenseNet121 ShuffleNetV2 MobileNetV2 |
| Scheduler | Cosine Annealing |
| Augmentation | RandomResizedCrop, Horizontal Flip |

Table 15: Hyperparameters for generating the distilled datasets.

| Setting | Learning Rate | $\eta$ |
|---|---|---|
| S1 | 0.001 | 1 |
| S2 | 0.001 | 2 |
| S3 | 0.0005 | 1 |
| S4 | 0.0005 | 2 |

Table 16: Hyperparameter settings with learning rate and $\eta$.

### G.2.1 CIFAR-100

This subsection outlines the hyperparameter configurations employed in the CIFAR-100 experiments, providing the necessary details to ensure reproducibility in future research.

**Pre-training phase.** Table 17 provides a comprehensive summary of the hyperparameters employed for training the models on the original CIFAR-100 dataset for generating the distilled dataset.

| Hyperparameters for Model Pre-training | |
|---|---|
| Optimizer | SGD |
| Learning Rate | 0.1 |
| Weight Decay | 1e-4 |
| Momentum | 0.9 |
| Batch Size | 128 |
| Epoch | 50 |
| Scheduler | Cosine Annealing |
| Augmentation | RandomCrop, Horizontal Flip |
| Loss Function | Cross-Entropy |

Table 17: Hyperparameters for CIFAR-100 Pre-trained Models.

**Evaluation Phase.** Table 18 outlines the hyperparameter configurations employed for the post-evaluation phase on the Distilled CIFAR-100 dataset.

### G.2.2 Tiny-ImageNet

This part describes the hyperparameter settings used in the Tiny-ImageNet experiments, offering comprehensive details to facilitate reproducibility for future studies.

| Hyperparameters for Post-Eval on R18, R50 and R101 | |
|---|---|
| Optimizer | AdamW |
| S1 | IPC1 (R50), IPC50 (R18,R50) |
| S2 | IPC10 (R18, R50) |
| S3 | IPC1 (R101), IPC10 (R101), IPC50 (R101) |
| S4 | IPC1 (R18) |
| Soft Label Generation | BSSL |
| Loss Function | KL-Divergence |
| Batch Size | 16 |
| Epochs | 1000 |
| Augmentation | RandomResizedCrop, Horizontal Flip, CutMix |

Table 18: Hyperparameters for post-evaluation task on ResNet18, ResNet50 and ResNet101 for CIFAR-100.

**Pre-training phase.** Table 19 presents a detailed overview of the hyperparameters used for model training on the original Tiny-ImageNet dataset.

| Hyperparameters for Model Pre-training | |
|---|---|
| Optimizer | SGD |
| Learning Rate | 0.1 |
| Weight Decay | 1e-4 |
| Momentum | 0.9 |
| Batch Size | 64 |
| Epoch | 150 |
| Scheduler | Cosine Annealing |
| Augmentation | RandomCrop, Horizontal Flip |
| Loss Function | Cross-Entropy |

Table 19: Hyperparameters for Tiny-ImageNet Pre-trained Models.

| Hyperparameters for Post-Eval on R18, R50 and R101 | |
|---|---|
| Optimizer | AdamW |
| S1 | IPC50 (R18) |
| S2 | IPC1 (R18) IPC10 (R18) |
| S3 | IPC50 (R50, R101) |
| S4 | IPC1 (R50, R101) IPC10 (R50, R101) |
| Soft Label Generation | BSSL |
| Loss Function | KL-Divergence |
| Batch Size | 16 |
| Epochs | 300 (IPC10, IPC50), 1000 (IPC1) |
| Augmentation | RandomResizedCrop, Horizontal Flip, CutMix |

Table 20: Hyperparameters for post-evaluation task on ResNet18, ResNet50 and ResNet101 for Tiny-ImageNet.

**Evaluation Phase.** Table 20 details the hyperparameter settings used during the post-evaluation phase on the Distilled Tiny-ImageNet dataset.

### G.2.3 ImageNette

This subsection describes the hyperparameter settings utilized in the ImageNette experiments, offering detailed information to facilitate reproducibility for subsequent studies.

**Pre-training phase.** Table 21 summarizes the hyperparameters used for training models on the original ImageNette dataset to generate the distilled dataset, ensuring clarity and reproducibility.

| Hyperparameters for Model Pre-training | |
|---|---|
| Optimizer | SGD |
| Learning Rate | 0.01 |
| Weight Decay | 1e-4 |
| Momentum | 0.9 |
| Batch Size | 128 |
| Epoch | 300 |
| Scheduler | Cosine Annealing |
| Augmentation | RandomReizeCrop, Horizontal Flip |
| Loss Function | Cross-Entropy |

Table 21: Hyperparameters for ImageNette Pre-trained Models.

**Evaluation Phase.** Table 22 details the hyperparameter settings applied during the post-evaluation phase on the Distilled ImageNette dataset.

| Hyperparameters for Post-Eval on R18, R50 and R101 | |
|---|---|
| Optimizer | AdamW |
| S2 | IPC50 (R101) |
| S3 | IPC10 (R18, R50) IPC50(R50) |
| S4 | IPC1(R18, R50, R101) IPC10 (R101) IPC50 (R18) |
| Soft Label Generation | BSSL |
| Loss Function | KL-Divergence |
| Batch Size | 16 |
| Epochs | 300 |
| Augmentation | RandomResizedCrop, Horizontal Flip, CutMix |

Table 22: Hyperparameters for post-evaluation task on ResNet18, ResNet50 and ResNet101 for ImageNette.

### G.2.4 ImageWoof

This section describes the hyperparameter settings used in the ImageWoof experiments, offering detailed information to facilitate reproducibility for future studies.

**Pre-training phase.** Table 23 presents a detailed overview of the hyperparameters utilized for training models on the original ImageWoof dataset to produce the distilled dataset.

**Evaluation Phase.** Table 24 presents the hyperparameter settings utilized during the post-evaluation stage on the distilled ImageWoof dataset, detailing the configurations applied for performance assessment.

### G.2.5 ImageNet-1K

This subsection outlines the hyperparameter configurations employed in the ImageNet-1K experiments, providing the necessary details to ensure reproducibility in future research.

| Hyperparameters for Model Pre-training | |
| --- | --- |
| Optimizer | SGD |
| Learning Rate | 0.1 |
| Weight Decay | 1e-4 |
| Momentum | 0.9 |
| Batch Size | 128 |
| Epoch | 50 |
| Scheduler | Cosine Annealing |
| Augmentation | RandomResizedCrop, Horizontal Flip |
| Loss Function | Cross-Entropy |

Table 23: Hyperparameters for ImageWoof Pre-trained Models.

| Hyperparameters for Post-Eval on R18, R50 and R101 | |
| --- | --- |
| Optimizer | AdamW |
| S1 | IPC1 (R101) |
| S2 | IPC50 (R18) |
| S3 | IPC10 (R18, R50) IPC50 (R50, R101) |
| S4 | IPC1 (R18, R50) IPC10 (R101) |
| Soft Label Generation | BSSL |
| Loss Function | KL-Divergence |
| Batch Size | 16 |
| Epochs | 300 |
| Augmentation | RandomResizedCrop, Horizontal Flip, CutMix |

Table 24: Hyperparameters for post-evaluation task on ResNet18, ResNet50 and ResNet101 for ImageWoof.

**Pre-training phase.** For ImageNet-1K, we employed the official PyTorch pretrained models, which have been extensively trained on the full ImageNet-1K dataset.

| Hyperparameters for Post-Eval on R18, R50 and R101 | |
| --- | --- |
| Optimizer | AdamW |
| S1 | IPC50 (R18, R50) |
| S2 | IPC1 (R18) IPC10 (R18, R50, R101) |
| S3 | IPC50 (R101) |
| S4 | IPC1 (R50, R101) |
| Soft Label Generation | BSSL |
| Loss Function | KL-Divergence |
| Batch Size | 16 |
| Epochs | 300 |
| Augmentation | RandomResizedCrop, Horizontal Flip, CutMix |

Table 25: Hyperparameters for post-evaluation task on ResNet18, ResNet50 and ResNet101 for ImageNet-1K.

**Evaluation Phase.** Table 25 provides a detailed overview of the hyperparameter settings used during the post-evaluation phase on the distilled ImageNet-1K dataset.

# H Additional Distilled Data Visualization

Additional visualizations of the distilled data generated by **FADRM** are provided in Fig. 7 (CIFAR-100), Fig. 9 (Tiny-ImageNet), Fig. 11 (ImageNette), Fig. 13 (ImageWoof), and Fig. 15 (ImageNet-1K). Furthermore, enhanced versions - **FADRM+** are presented in Fig. 8 (CIFAR-100), Fig. 10 (Tiny-ImageNet), Fig. 12 (ImageNette), Fig. 14 (ImageWoof), and Fig. 16 (ImageNet-1K).

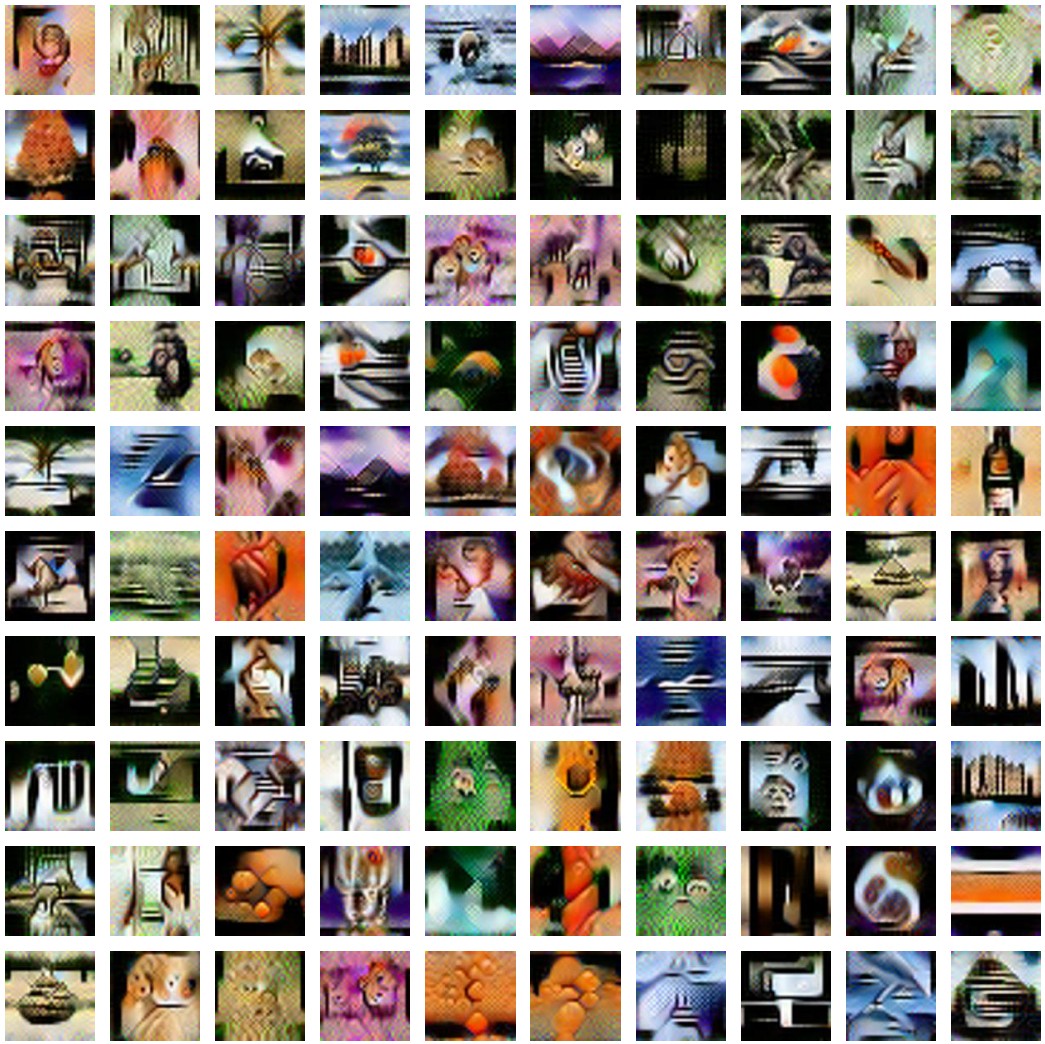

Figure 7: Visualization of synthetic data on CIFAR-100 generated by **FADRM**.

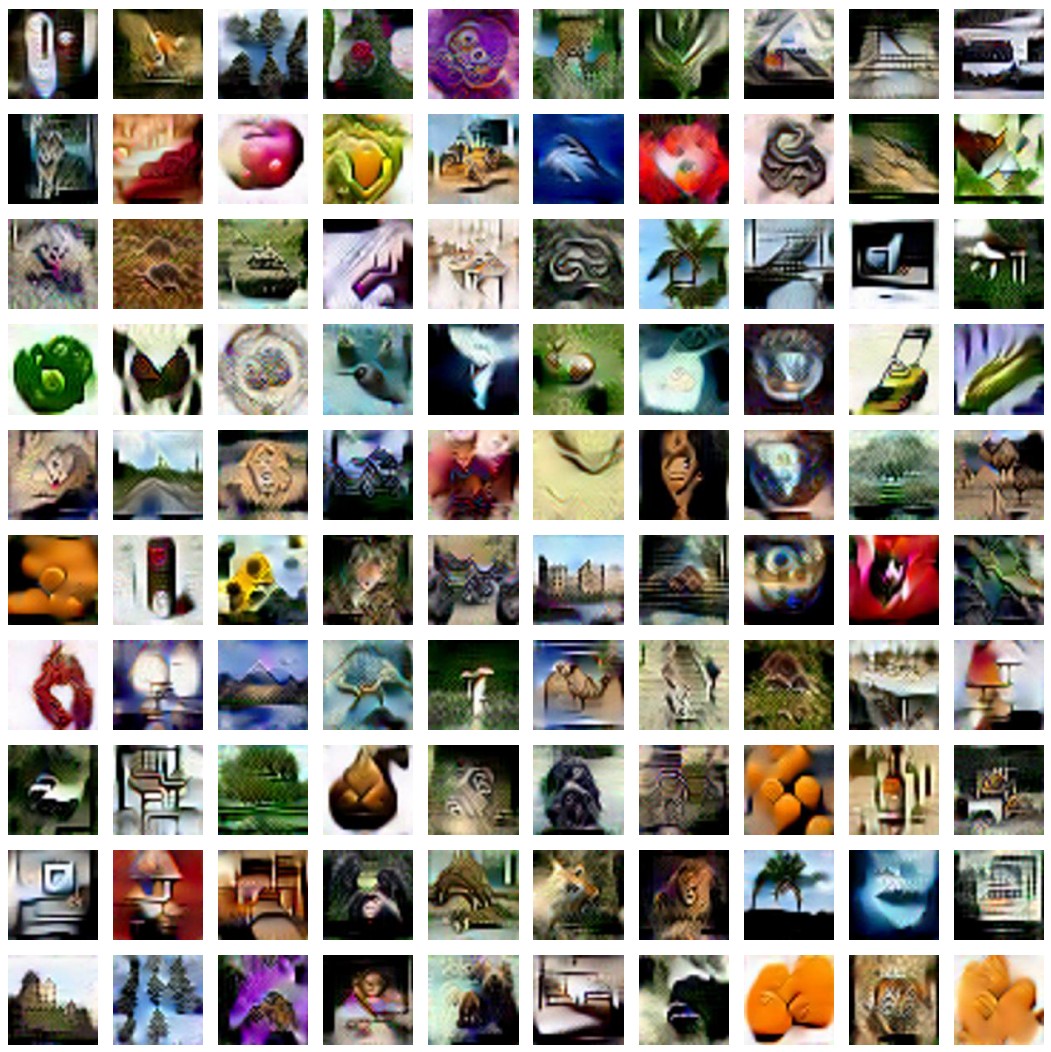

Figure 8: Visualization of synthetic data on CIFAR-100 generated by **FADRM+**.

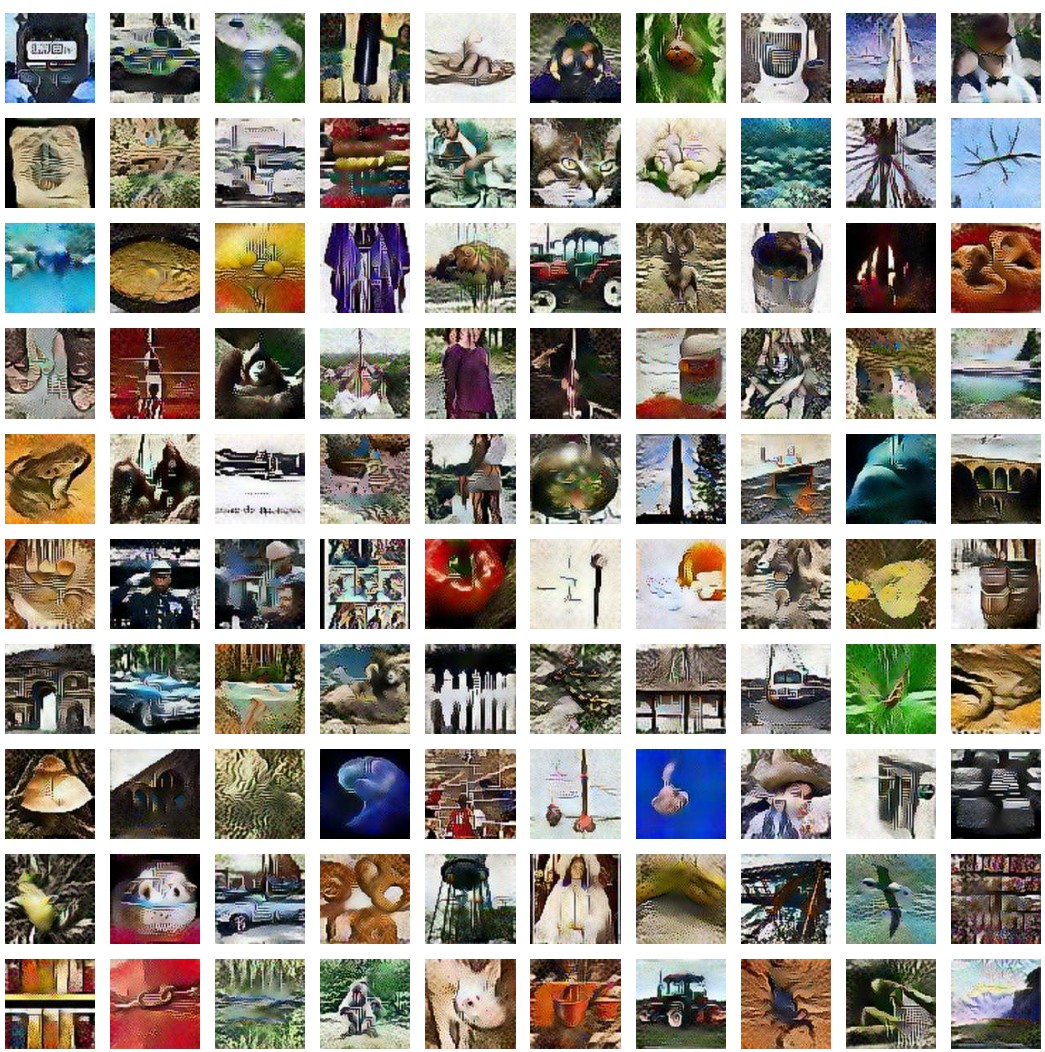

Figure 9: Visualization of synthetic data on Tiny-ImageNet generated by **FADRM**.

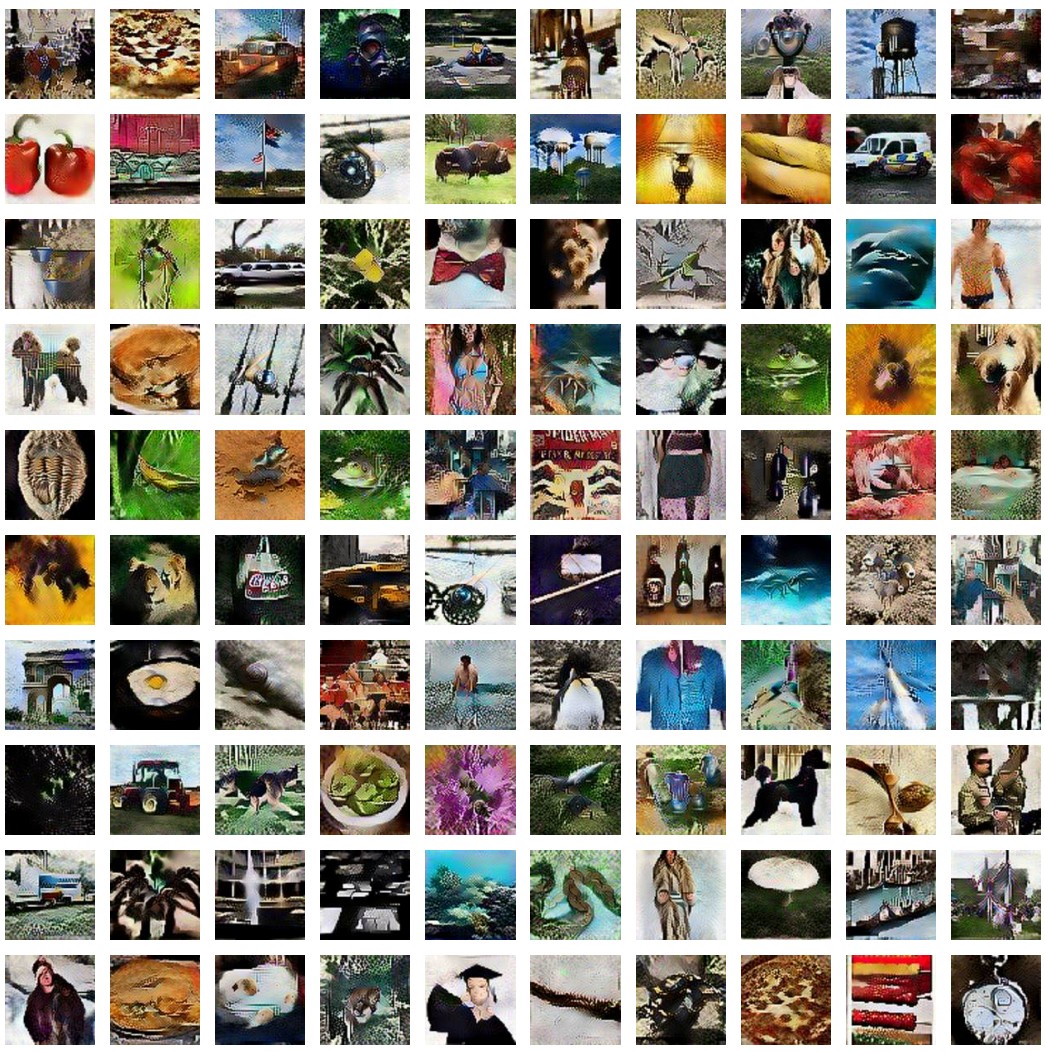

Figure 10: Visualization of synthetic data on Tiny-ImageNet generated by **FADRM+**.

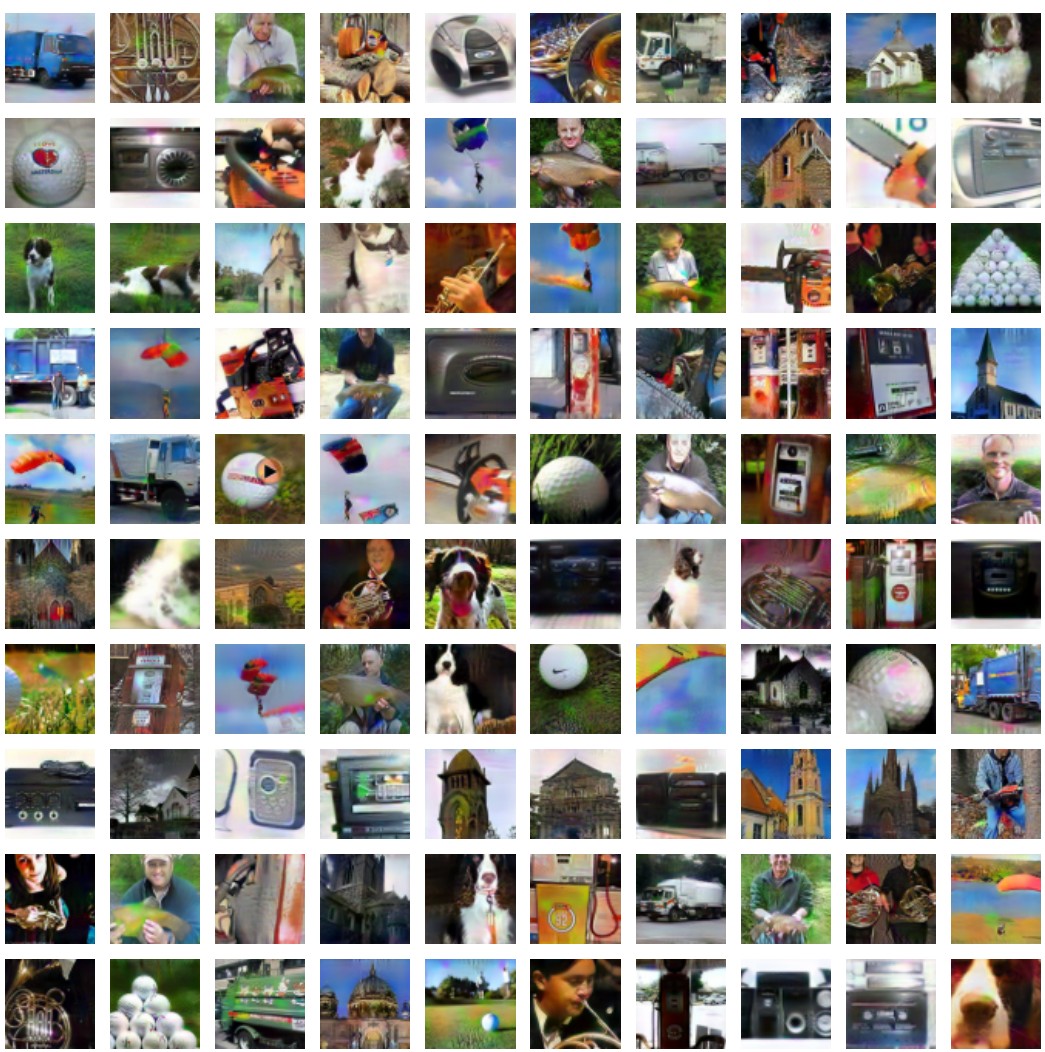

Figure 11: Visualization of synthetic data on ImageNette generated by **FADRM**.

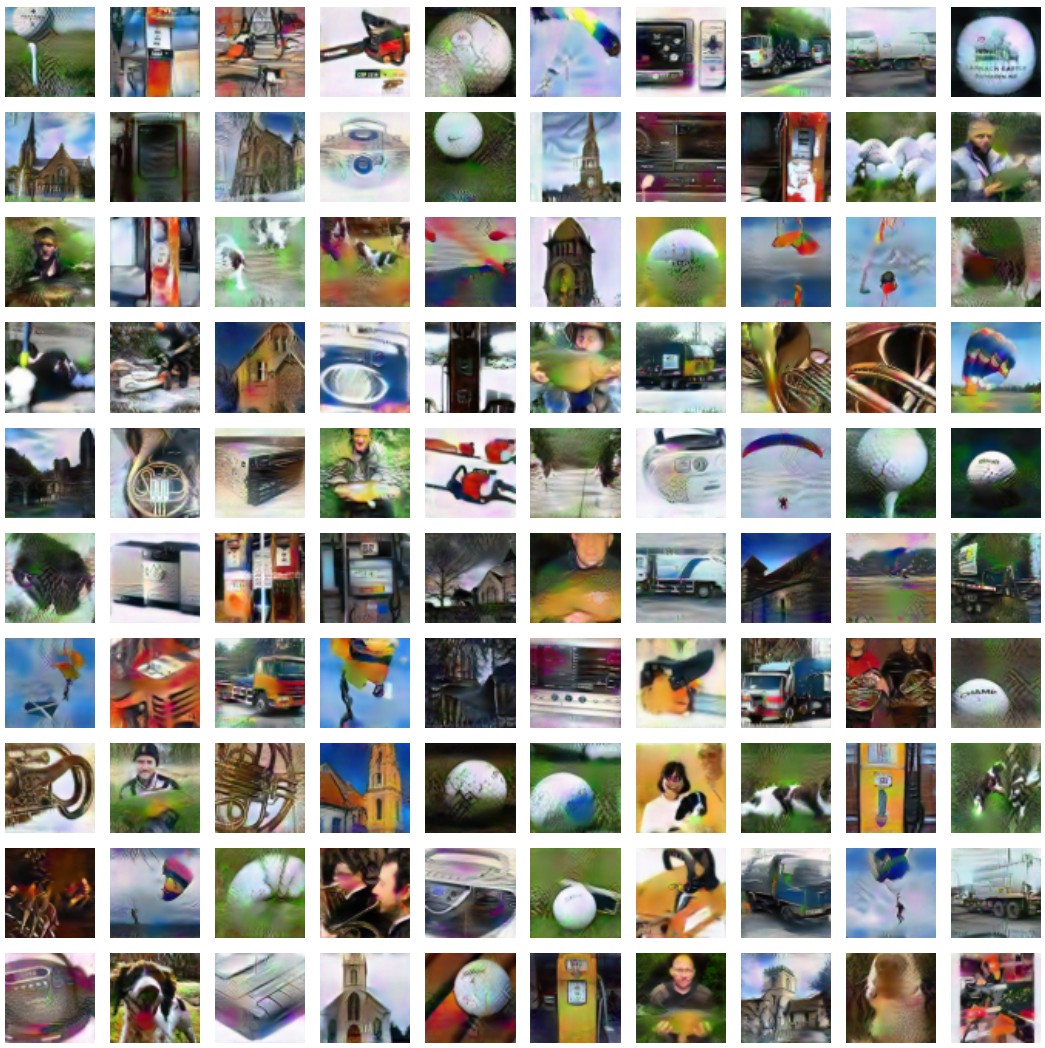

Figure 12: Visualization of synthetic data on ImageNette generated by **FADRM+**.

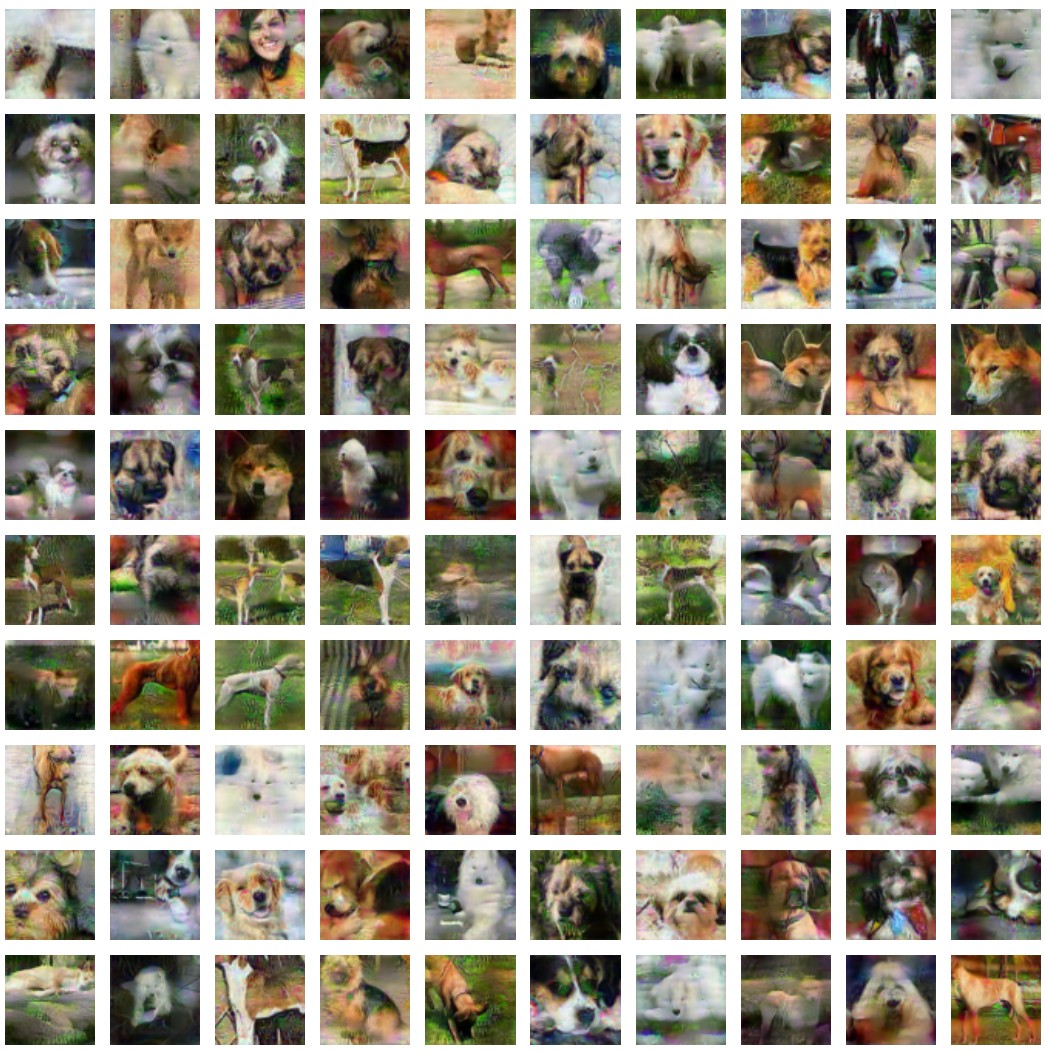

Figure 13: Visualization of synthetic data on ImageWoof generated by **FADRM**.

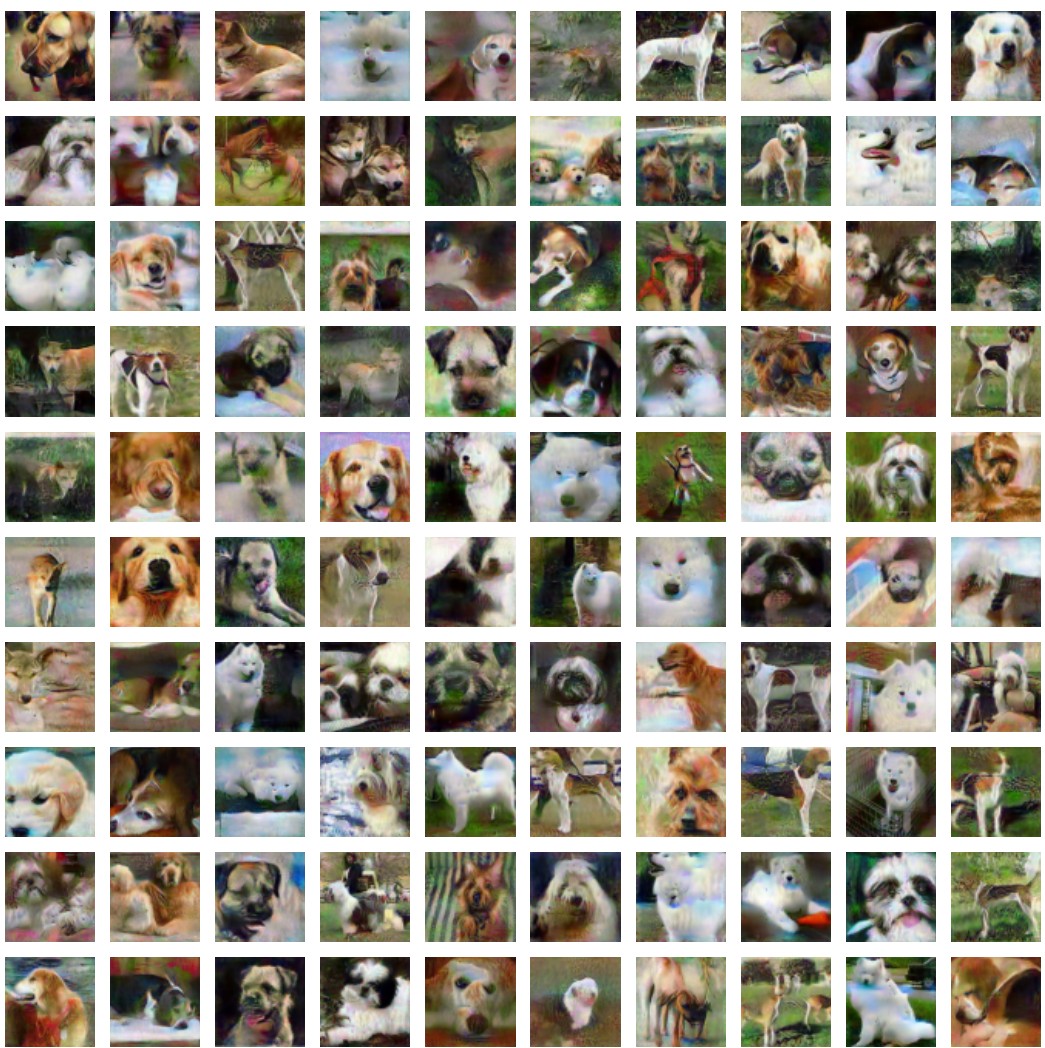

Figure 14: Visualization of synthetic data on ImageWoof generated by **FADRM+**.

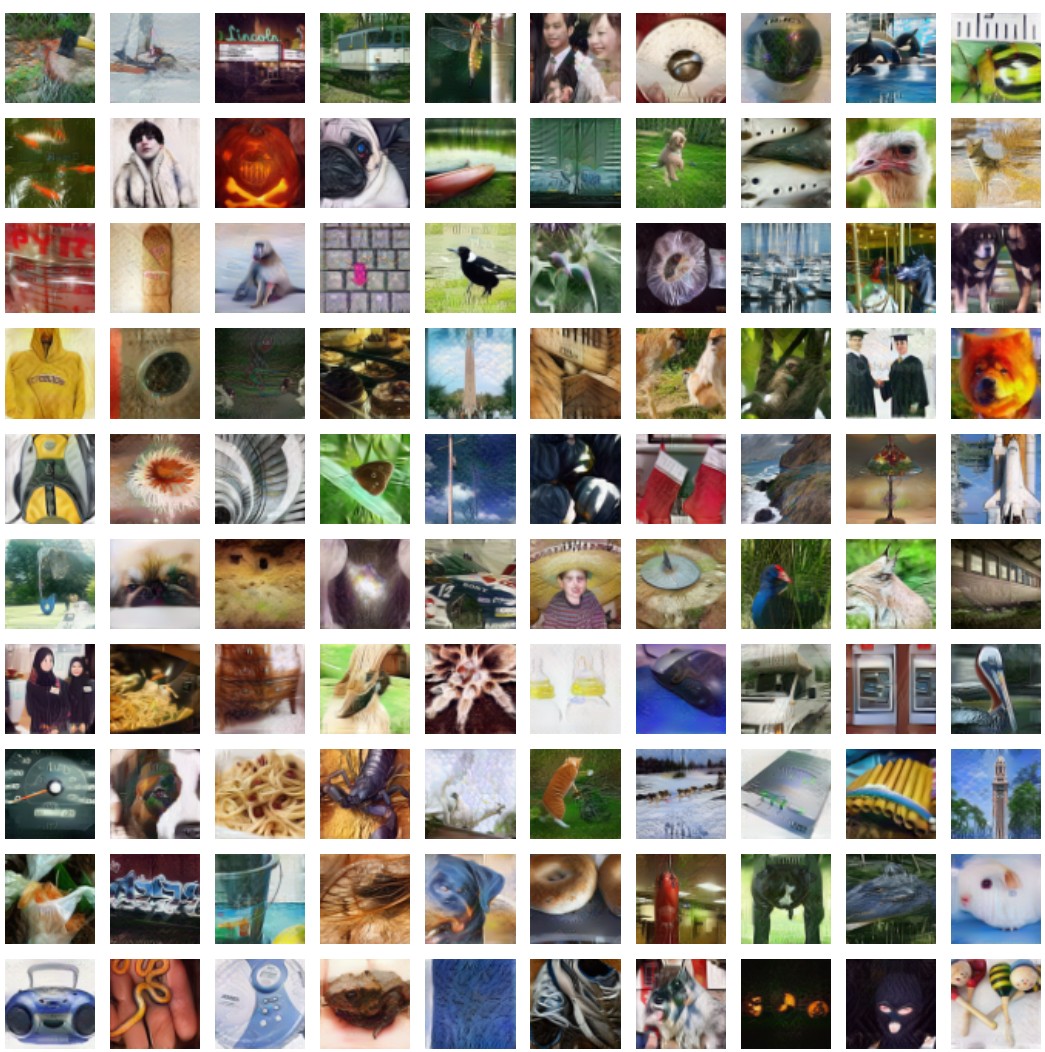

Figure 15: Visualization of synthetic data on ImageNet-1K generated by **FADRM**.

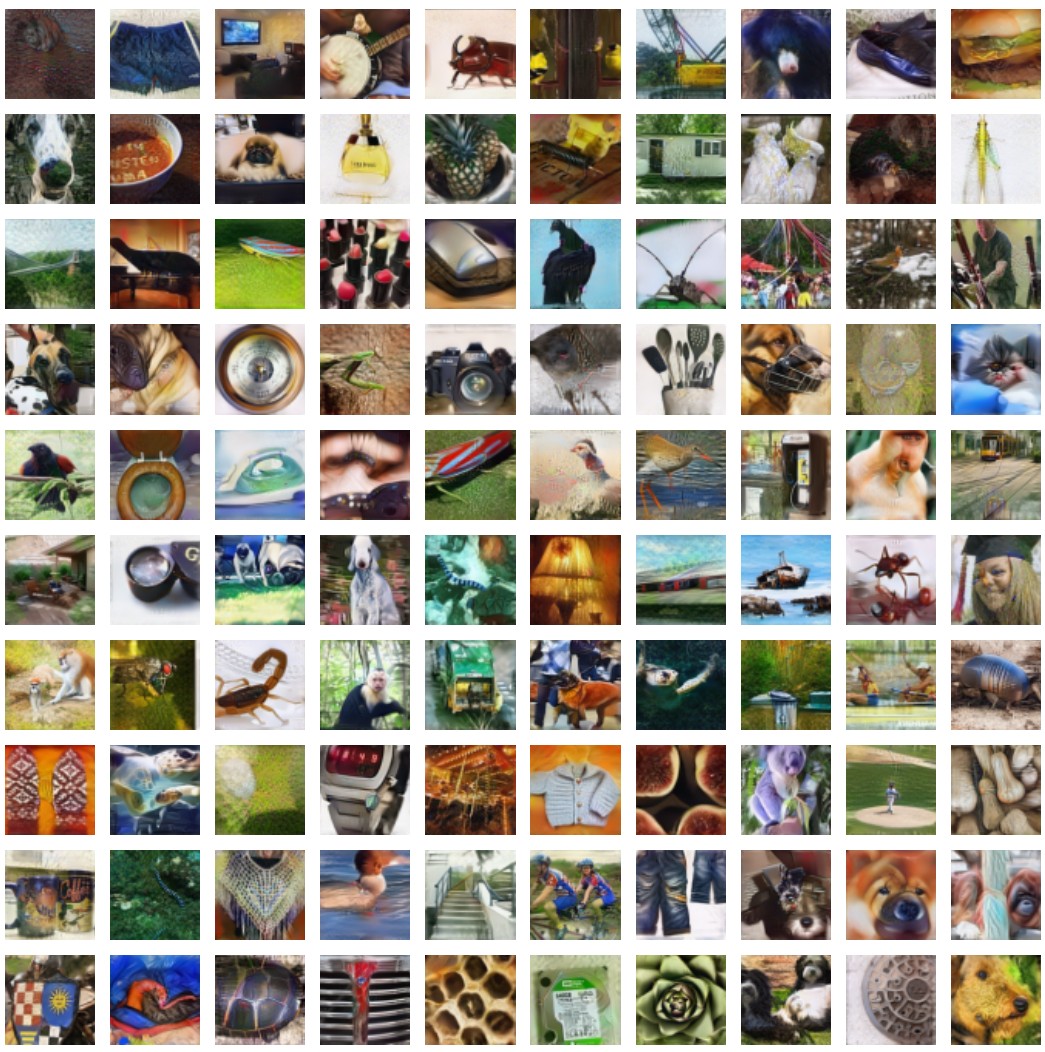

Figure 16: Visualization of synthetic data on ImageNet-1K generated by **FADRM+**.

