# OpenReview forum: "FADRM: Fast and Accurate Data Residual Matching for Dataset Distillation"
_NeurIPS.cc/2025/Conference — NeurIPS 2025 poster_

### Official Review · Reviewer_kJwL · 2025-06-30

**Clarity:** 3
**Significance:** 2
**Originality:** 3
**Rating:** 2
**Confidence:** 4

**Summary:**

This paper proposes FADRM that introduces data-level residual connections to preserve important information during synthetic data generation.
The FADRM combines multi-resolution optimization and adjustable residual merging to improve both efficiency and accuracy.

**Questions:**

See Weaknesses.

**Ethical Concerns:**

["NO or VERY MINOR ethics concerns only"]

**Final Justification:**

I remain concerned about the novelty of this work, especially in terms of multi-resolution optimization, skip connections, and mixed-precision training, which are established techniques in the field. Reviewer Ne6j also noted this point. Therefore, I believe the manuscript requires further development before it can be considered suitable for publication.

**Limitations:**

Yes

**Quality:**

2

**Strengths And Weaknesses:**

Strengths:

**S1**: The extension of the residual mechanism from model-level to data-level is conceptually novel.

**S2**: The method achieves impressive performance across various datasets and architectures.


Weaknesses:

**W1**:
Figure 2 is not clear.
Whether every **Data Residual Block** includes a **Recover**?
Why **Downsample** upsize the image size in **Resampler**, and vice versa?
This increases the difficulty of understanding.

**W2**:
This paper claims FADRM achieves Fast and Accurate matching.
RDED is a relevant and efficient baseline, which is missing from Figure 1 and Table 2.

**W3**:
How does the author explain the observation in Figure 3, where information vanishing still occurs even with ARC? Furthermore, why is there such a big shock compared to without ARC? This raises questions about whether ARC introduces instability into the distillation process. These phenomena should be discussed in detail.

**W4**:
Theorem 2 relies on the assumption in Eq.(8), which is crucial for concluding that FADRM generalizes better.
However, the paper does not provide empirical validation or intuitive justification for when this assumption holds.
Thus, the theoretical advantage is conditional and its practical applicability is unclear.
The paper reports accuracy and runtime but never measures either side of Eq. (8), such as empirical Rademacher estimates, margin distributions or PAC-Bayes surrogates.

---

> ### Author Rebuttal · Authors · 2025-07-30
>
> We sincerely thank you for your constructive comments. We are encouraged that you found our method novel, and appreciate your recognition of its performance and efficiency improvements. We would like to address the comments and questions below.
>
>
> > W1: Figure 2 is not clear.
>
> We sincerely thank the reviewer for the thoughtful question. To clarify, each Data Residual Block does indeed include a Recover step. We also appreciate the reviewer for pointing out the labeling issue in Figure 2, upon review, we confirm that the downsample and upsample labels were inadvertently swapped, which may have caused confusion. We will correct the labels and revise the figure to improve clarity in the final version.
>
>
> > W2: Comparison to non-optimization based methods.
>
> We thank the reviewer for the question. RDED is a non-optimization-based framework that selects representative patches using a pretrained model, without involving any further image-level optimization. In contrast, our method is optimization-based and involves iterative refinement of synthetic images, which inevitably incurs higher computational cost. Nonetheless, such optimization typically leads to superior performance over non-optimization approaches.
>
> To ensure a fair comparison of efficiency, we report results against optimization-based baselines that involve similar iterative procedures and are scalable to large-scale datasets such as ImageNet-1K. As a result, RDED is not included in Table 2.
>
> > W3: Further explanation of Information Vanishing.
>
> We thank the reviewer for the question. First, we would like to clarify that our goal is not to eliminate information vanishing entirely, but to mitigate it. As optimization progresses, fine-grained local details are gradually suppressed in favor of dominant global structures. Once global information becomes saturated, continued suppression of local details inevitably leads to information vanishing.
>
> The sharp surge observed in Figure 3 occurs as ARC injects rich fine-grained details back into the data. This sudden reintroduction of diverse semantic content naturally leads to a spike in feature-level entropy. Importantly, this surge reflects the restoration of previously suppressed local details, rather than instability or noise.
>
> To verify that ARC preserves semantic consistency rather than introducing disruptive information, we conduct two complementary analyses. First, we measure the **cosine similarity of features** before and after ARC injection across three ARC stages ($k=3$). Second, we evaluate the **accuracy of the distilled images** before and after ARC using a pretrained model as a verifier.
>
> As shown in the table below, feature-level cosine similarity remains high (≥ 0.88) across all stages, indicating that the semantic structure is well preserved. While verifier accuracy drops slightly (from 100% to 97%) at the first injection stage, it quickly recovers in later stages, suggesting that ARC introduces **semantically aligned residual content** rather than harmful noise. These results demonstrate that the observed entropy increase stems from meaningful details, not instability in the distillation process.
>
> |     Stages                    |  Feature Cosine Similarity   |   Before ARC Accuracy  |   After ARC Accuracy  |
> |:---------------------------:|:----------------------------:|:----------------------:|:---------------------:|
> | Residual Injection Stage #1 |           0.89              |         100%           |         97%           |
> | Residual Injection Stage #2 |           0.88              |         100%           |         100%          |
> | Residual Injection Stage #3 |           0.88              |         100%           |         100%          |
>
>
> > W4: More analysis on assumption 8.
>
> We thank the reviewer for the insightful suggestion. We first elaborate on the intuition behind Assumption 8 and then empirically verify whether it generally holds. Starting from Assumption 8 and rearranging the right-hand side, we obtain:
>
> $$
> \mathfrak{R}\_n(\mathcal{H} \circ \tilde{\mathcal{C}}^{\text{res}}) - \mathfrak{R}\_n(\mathcal{H} \circ \mathcal{O}) > \left( \frac{L\_h L\_\ell}{2B} + \frac{L\_h}{2} \right) \cdot \Delta,
> $$
>
> where $\mathcal{O}$ denotes the original data patch, and $\tilde{\mathcal{C}}^{\text{res}}$ refers to the class induced by the optimized version of the original patch.
>
> Since both $L\_\ell$ and $L\_h$ are small due to the use of high-temperature KL loss and strong regularization (e.g., CutMix), the coefficient $\left( \frac{L\_h L\_\ell}{2B} + \frac{L\_h}{2} \right)$ is sufficiently small. As a result, the right-hand side becomes negligible in our setting and can be conservatively omitted to simplify the analysis, which is:
>
> $$
> \mathfrak{R}\_n(\mathcal{H} \circ \tilde{\mathcal{C}}^{\text{res}}) > \mathfrak{R}\_n(\mathcal{H} \circ \mathcal{O}),
> $$
>
> The intuitions is that if the Rademacher complexity of the optimized patches exceed that of the original ones, theorem 2 typically holds.
>
> To validate if this inequality holds empirically, we estimate the **empirical Rademacher complexity**, defined as:
>
> $$
> \widehat{\mathfrak{R}}\_n(\mathcal{H} \circ \mathcal{S}) = \mathbb{E}\_{\sigma} \left[ \sup\_{h \in \mathcal{H}} \frac{1}{n} \sum\_{i=1}^n \sigma\_i h(x\_i) \right],
> $$
>
> where $\sigma\_i$ are i.i.d. Rademacher variables from $\{-1, +1\}$ and $\mathcal{S}$ is the dataset. Although computing the population complexity is intractable, it is well known that:
>
> $$
> \lim\_{n \to \infty} \widehat{\mathfrak{R}}\_n(\mathcal{H}) = \mathfrak{R}\_n(\mathcal{H}),
> $$
>
> so the empirical proxy becomes more accurate as $n$ increases.
>
> In practice, we approximate this quantity via *random label memorization* on a IPC=50 dataset (n=50,000). A standard empirical estimate only involves binary classification, which we found to be too simple and less discriminative in our setting. To address this, we increase task difficulty by assigning **random labels uniformly from a large label space** (e.g., 500 and 1000 classes), making the memorization task substantially harder.
>
> Concretely, for a given dataset $\mathcal{D} = \{x\_i\}\_{i=1}^n$, we construct a randomized label version $\mathcal{D}\_{\text{rand}} = \{(x\_i, y\_i^\text{rand})\}\_{i=1}^n$ by sampling:
>
> $$
> y\_i^\text{rand} \sim \mathcal{U}\{1, 2, \dots, K\}, \quad \text{where } K \in \{500, 1000\}.
> $$
>
> We apply this procedure to both the original patch set and its optimized counterpart, and train three distinct models on each using cross-entropy loss. The minimum training loss across runs is reported as a proxy for Rademacher complexity: lower loss indicates higher capacity to fit random labels, and thus higher empirical complexity.
>
> Across all settings, whether under moderate (500-way) or challenging (1000-way) classification tasks, we consistently observe that the optimized data achieves **lower training loss** than the original. This suggests that the optimized data is easier to fit, reflecting a higher capacity of the hypothesis class to memorize random labels, thereby indicating a higher empirical Rademacher complexity.
>
> |  Models         | $\mathcal{O}$ loss | $\tilde{C}\_{res}$ loss |
> |:-------------------------:|:----------:|:------:|
> |  ResNet-18 (500-way)      |   0.0022   |  0.0016 |
> |  ResNet-18 (1000-way)     |   0.0034   |  0.0026 |
> |  MobileNet-V2 (500-way)   |   0.0016   |  0.0004 |
> |  MobileNet-V2 (1000-way)  |   0.0019   |  0.0006 |
> |  ShuffleNet-V2 (500-way)  |   0.0457   |  0.0414 |
> |  ShuffleNet-V2 (1000-way) |   0.0309   |  0.0234 |
>
> These findings offer strong empirical support for Assumption 8, suggesting that optimized patches indeed yield *higher empirical Rademacher complexity*. Therefore, the theorem 2 becomes applicable to real-world scenarios.

---

> ### Author Response · Authors · 2025-08-05
>
> We sincerely thank the reviewer again for the follow-up. We respectfully disagree with several points and would like to clarify them further to avoid potential misunderstandings.
>
> > (1) RDED comparison and fairness
>
> Our main comparison focuses on optimization-based methods, as FADRM involves iterative refinement while RDED does not. For completeness, we include RDED below:
>
> | Method | Efficiency (s/sample) | Accuracy (%) |
> |--------|------------------------|--------------|
> | RDED   | 0.0399                 | 42.0         |
> | FADRM  | 0.47                   | **47.7**     |
>
> FADRM is slower due to its optimization steps but achieves a **+5.7% accuracy gain**, which we believe is a meaningful trade-off. This result will be included in the revision.
>
>
> > (2) Effectiveness of ARC.
>
> We respectfully disagree with the claims that "continued optimization could lead to convergence with the W/O ARC curve" and that the "final trends are the same," as these are not supported by the trends in **Figure 3**.
>
> Applying ARC at **iteration 3500** yields **higher information density after 500 iterations** than applying it at **iteration 3000** with the same budget (see w/ ARC at 3400 vs. 4000 iterations). This indicates that **longer optimization with ARC preserves and stabilizes information at a higher plateau**. In contrast, the W/O ARC curve exhibits a **steady decline** throughout optimization, reflecting continuous information loss, making it fundamentally impossible for the two curves to converge.
>
> To further refute the reviewer’s suggestion that ARC’s improvement is merely temporary, we present the table below (also included in our response to reviewer ccas Weakness 4), which shows **consistent accuracy improvements with ARC** across both shallow and deep optimization stages. This directly contradicts the notion that ARC’s effect diminishes over time.
>
> | Iterations | FADRM w/ ARC (%) | FADRM w/o ARC Accuracy (%) |
> |:----------:|:------------------:|:--------------------:|
> | 500        |    40.0            |    39.7              |
> | 1000       |    45.2            |    44.8              |
> | 1500       |    47.2            |    46.5              |
> | 2000       |    47.7            |    46.1              |
> | 2500       |    48.0            |    46.5              |
> | 3000       |    48.0            |    45.4              |
> | 3500       |    48.2            |    44.5              |
> | 4000       |    48.4            |    43.1              |
>
> > (3) Technical novelty.
>
> We respectfully note that the key components of our method are not drawn from existing distillation techniques. We introduce **data-level residual connections**, which reinterpret skip connections for synthetic data refinement, bridging the gap between model optimization and data synthesis.
>
> We also **shift the focus to optimization-level efficiency**, moving beyond model size or initialization to improve efficiency by identifying and reducing redundant computation within the optimization process. To our knowledge, this perspective has not been explored in prior uni-level framework.

---

> > ### Comment · Reviewer_kJwL · 2025-08-09
> >
> > I appreciate the authors’ detailed responses and clarifications. However, I remain concerned about the novelty of this work, especially in terms of multi-resolution optimization, skip connections, and mixed-precision training, which are established techniques in the field. Reviewer Ne6j also noted this point. Therefore, I believe the manuscript requires further development before it can be considered suitable for publication.

---

> ### Author Response · Authors · 2025-08-09
>
> We sincerely thank the reviewer for the response, and we are pleased that our earlier discussion has addressed your concerns regarding the effectiveness of ARC.
>
> We would like to clarify that Reviewer Ne6j has consistently recognized the novelty of our main contribution ARC (data-level residual connection). After we explained that MPT (mixed precision training) and MRO (multi resolution optimization) are designed to shift the focus toward optimization-level refinement, a direction that has been consistently overlooked in prior uni-level frameworks, Ne6j agreed this also addressed their concern regarding the novelty.
>
> We respectfully note that, to the best of our knowledge, ours is the first work to explicitly identify the issue of information vanishing and to apply data-level residual connections as a solution. We also note that the novelty of our work has been clearly recognized by all other reviewers.

---

### Official Review · Reviewer_Ne6j · 2025-06-30

**Clarity:** 3
**Significance:** 2
**Originality:** 3
**Rating:** 4
**Confidence:** 4

**Summary:**

This paper identifies the information vanishing problem in uni-level frameworks and addresses it with Adjustable Residual Connection (ARC), which applies the idea of model-level skip connections at the data level. It also introduces Multi-Resolution Optimization (MRO), which distills datasets at lower resolutions to improve training efficiency. Mixed Precision Training (MPT) is used to further reduce training time and memory use. The proposed method, FADRM, achieves state-of-the-art results in both accuracy and efficiency across multiple benchmarks.

**Questions:**

Could the authors clarify why the evaluation criteria used for time efficiency comparisons differ between Table 2 (left) and Table 3?
Lines 289-291 indicate that MRO reduces peak memory usage. However, Table 3 shows that MPT alone reduces peak memory to 2.9GB. Could the authors clarify whether the observed memory reduction is indeed due to MRO rather than MPT?

**Ethical Concerns:**

["NO or VERY MINOR ethics concerns only"]

**Final Justification:**

The authors’ clarifications, especially on efficiency, convinced me that MPT and MRO, though not major novelties individually, play meaningful roles in improving overall efficiency. The thorough responses have addressed my main concerns, leading to a borderline accept.

**Limitations:**

yes

**Quality:**

2

**Strengths And Weaknesses:**

Strength

(+) The paper provides a clear theoretical justification and empirical validation for the phenomenon of information vanishing in uni-level frameworks, effectively defining and analyzing this critical issue.

(+) The authors present an original approach by interpreting model-level skip connections as data-level skip connections, successfully preserving critical information from the original dataset during the distillation process.

Weakness

(-) It remains unclear whether the reported improvements in computational efficiency attributed to MRO are genuinely derived from this approach itself, or if they primarily result from MPT.

(-) Although Table 3 demonstrates notable performance gains when comparing FADRM to SRe^2L++, the reported efficiency and GPU memory are same. Thus, it is questionable whether FADRM truly offers substantial efficiency enhancements over SRe^2L++.

(-) Table 1 indicates minor performance gains for ImageNet-1K with ResNet-18. However, without reported statistical variability from multiple runs, commonly provided in other baselines, assessing the statistical significance of these improvements and making a rigorous comparison with baselines is difficult.

---

> ### Author Rebuttal · Authors · 2025-07-29
>
> We sincerely thank you for your encouraging feedback. We appreciate your recognition of the clear theoretical justification and the novelty of our data-level residual design. We address the remaining concerns in the detailed responses below.
>
> > W1: Contribution of individual componenets on efficiency gains.
>
> Thank you for the suggestion. To clarify the individual contributions of Mixed Precision Training (MPT) and Multi-Resolution Optimization (MRO) to computational efficiency, we conducted an ablation study as shown in the table below, the experiment is conducted on ImageNet-1k, where efficiency is measured based on the time needed for generating a single distilled image.
>
> | Settings           | Time Cost (s/img) | Minimum GPU Usage (GB) | Peak GPU Usage (GB)  |
> |:------------------:|:------------------:|:----------------------:|:--------------------:|
> | w/o MPT, w/o MRO   | 1.26               | 5.3                    | 5.3                  |
> | w/o MPT, w/ MRO    | 0.84               | 4.0                    | 5.3                  |
> | w/ MPT, w/o MRO    | 0.52               | 2.9                    | 2.9                  |
> | w/ MPT, w/ MRO     | 0.47               | 2.5                    | 2.9                  |
>
> As shown, MRO is primarily intended to reduce the overall wall-clock time required to optimize a single image by iteratively performing training on downsampled versions. Since the process still involves full-resolution updates, the peak GPU usage remains unchanged. In contrast, MPT is designed to reduce both computation time and peak memory consumption by leveraging lower-precision operations throughout.
>
> While MPT accounts for the majority of the efficiency improvement, MRO provides complementary gains. For instance, when comparing the configurations with and without MRO under MPT, we observe a reduction of 0.05 seconds per image. Although this difference may appear marginal at the individual level, it becomes meaningful when scaled. For example, generating a 50 IPC ImageNet-1K dataset (i.e., 50,000 images) results in:
>
> $$
> 50{,}000 \times 0.05\ \text{seconds} = 2{,}500\ \text{seconds} \approx 42\ \text{minutes}
> $$
>
> In summary, MPT is the primary contributor to efficiency, and MRO further improves performance by iteratively optimze the images on a smaller resolution.
>
>
> > W2: Efficiency enhancements over SRe$^2$L++.
>
> Thank you for the question. We clarify that SRe$^2$L++ does not adopt Mixed Precision Training (MPT) or Multi-Resolution Optimization (MRO), and requires 4000 iterations to optimize each image. In contrast, our proposed method FADRM applies both MPT and MRO, and completes optimization in only 2000 iterations. This results in substantial efficiency gains, as shown in Table 2 (left), including a reduction of approximately 2.05 seconds in wall-clock time per image and a significant decrease in peak GPU memory usage.
>
> To illustrate the impact at scale, generating a 50 IPC ImageNet-1K distilled dataset (50,000 images) would lead to:
>
> $$
> 50{,}000 \times 2.05\ \text{seconds} = 102{,}500\ \text{seconds} \approx 28.5\ \text{hours}
> $$
>
> This demonstrates the practical advantage of FADRM over SRe$^2$L++ in large-scale settings. Additionally, we clarify that Table 3 serves a different purpose: it evaluates the impact of applying MPT across various methods and confirms that MPT does not degrade the quality of the distilled dataset. The analysis is not limited to FADRM and is intended to show the broader applicability of MPT in the field of dataset distillation.
>
> > W3: Clarification for performance gains.
>
> We thank the reviewer for raising this important point.
>
> Firstly, we would like to clarify that the performance gains in the ImageNet-1K ResNet-18 setting are substantial. For single-model distillation, our method FADRM achieves clear improvements over the strongest baseline RDED. For example, at IPC = 10, FADRM outperforms RDED by +5.7%, which we believe is both meaningful and robust. For multi-model distillation, FADRM+ also consistently surpasses other approaches across all IPC levels, as reported in Table 1. The only case of relatively moderate improvement is at IPC = 50, where FADRM slightly outperforms CV-DD by +0.6%.
>
> To confirm the consistency and statistical reliability of our results, we have conducted additional runs and report the mean and standard deviation across 3 independent trials on ImageNet-1K with ResNet-18, as shown in the table below (CV-DD didn't report statistical variations). The results further demonstrate the stability and robustness of FADRM across runs.
>
> | IPC | RDED (single) | FADRM (single) | CV-DD (multi) | EDC(multi) | FADRM+(multi) |
> |:-----------------:|:----------:|:----------:|:----------:|:----------:|:---------------:|
> | 1        | 6.6 $\pm$ 0.2    |  7.1 $\pm$ 0.3    | 9.2     | 12.8 $\pm$ 0.1  | **14.3** $\pm$ 0.2 |
> | 10       | 42.0 $\pm$ 0.1   |  47.8 $\pm$ 0.2   | 46.0    | 48.6 $\pm$ 0.3  | **50.4** $\pm$ 0.1|
> | 50       | 56.5 $\pm$ 0.1   |  59.9 $\pm$ 0.2   | 59.5    | 58.0 $\pm$ 0.2  | **60.4**$\pm$ 0.3 |
>
> > Q1: Clarification on evaluation criteria of efficiency.
>
> Thank you for the question. There are two distinct evaluation criteria used to assess time efficiency in the paper: (1) seconds per image generation, as shown in Table 2 (left), and (2) milliseconds per image update, as shown in Table 3. The choice of metric depends on the evaluation objective.
>
> In Table 2 (left), we report seconds per image generation, which captures the total wall-clock time required to generate a single distilled image. This metric accounts for various factors, including the total number of optimization iterations, the varying update time across different image resolutions introduced by MRO, and additional overhead such as loading precomputed statistics in EDC. We choose this metric because, under the MRO setting, image resolution changes throughout the optimization process, making it difficult to fairly measure efficiency using a single per-update timing. In contrast, seconds per image generation provides a more comprehensive and resolution-agnostic view of overall efficiency.
>
> In contrast, Table 3 focuses on optimization-level efficiency under fixed-resolution settings, where only MPT is applied and MRO is excluded. In this case, we use milliseconds per image update to isolate and assess the efficiency gains specifically attributable to MPT.
>
> In summary, we adopt different metrics to match the evaluation focus: Table 2 reflects overall generation efficiency, accounting for all contributing factors, while Table 3 isolates the per-iteration benefits of MPT without the confounding effects introduced by varying image resolutions.
>
>
> > Q2: Clarification on Peak Memory Usage.
>
> Thank you for the question. We would like to clarify that in line 289, the phrase "optimization-level refinement and MRO" refers collectively to two efficiency enhancing components used in our method: optimization-level refinement corresponds to Mixed Precision Training (MPT), while MRO refers to Multi-Resolution Optimization. The sentence aims to highlight that the integration of both techniques leads to improved efficiency.
>
> To answer the question directly: the observed reduction in peak GPU memory usage is primarily attributed to MPT, which reduces memory consumption by employing lower-precision arithmetic during optimization. In contrast, MRO is designed to improve overall training efficiency by reducing computation time, as discussed in our response to Weakness 1.
>
> We hope this clarifies that the efficiency gains mentioned in line 289 result from the combined use of MPT and MRO, with MPT responsible for the peak memory reduction.

---

### Official Review · Reviewer_Ujgz · 2025-06-30

**Clarity:** 1
**Significance:** 3
**Originality:** 3
**Rating:** 4
**Confidence:** 4

**Summary:**

This paper introduces a novel data residual matching strategy for database distillation, aiming to significantly reduce training time and memory usage. The approach demonstrates substantial improvements in both computational efficiency and model performance, reducing resource demands by 50% compared to baseline methods.

**Questions:**

**(Q1)** What task was concerned here for performance evaluation, and why was it chosen? Was it image classification?\
**(Q2)** What models were trained using the distilled datasets? Were the training configurations (e.g., hyperparameters, epochs) kept consistent across all methods/models?

**Ethical Concerns:**

["NO or VERY MINOR ethics concerns only"]

**Final Justification:**

My concerns have been properly addressed by the authors, and after reviewing the comments from other reviewers, I have decided to raise my rating from 3 to 4.

**Limitations:**

The paper does not clearly discuss its limitations. In particular, the generalizability of the method to architectures beyond ResNet is a concern that should be addressed, especially when the proposed approach may rely on residual connections.

**Quality:**

3

**Strengths And Weaknesses:**

**Strengths**

**(S1)** The proposed method yields clear performance gains over baseline distillation techniques.\
**(S2)** Training time and memory consumption are significantly reduced, making the approach highly practical for real-world applications where efficiency is crucial.


**Weaknesses**

**(W1)** There seems a labelling error in Figure 2, where the “downsample” and “upsample” lables in the resampler module appear to be mistakenly swapped.\
**(W2)** The evaluated task is not clearly stated. While it appears to be image classification, this must be explicitly mentioned for clarity.\
**(W3, Major)** The models used for training and evaluation are not clearly described. Table 1 suggests the use of ResNet-based architectures, but the paper should clearly state which models were used and whether the approach generalizes to non-ResNet architectures. It is unclear whether the selection here is related to the design of the residual blocks in the proposed framework. This lack of clarity raises a major concern regarding the paper’s methodology.\
**(W4 - Minor)** Figure 2 could be misleading, as it shows an equal number of patches for the original and distilled datasets. Typically, the distilled set should contain significantly fewer samples.\
**(W5)** The peak memory results for FADRM+ in Tables 2 and 4 differ, while those for the other models remain consistent. Could you clarify the reason for this inconsistency?

---

> ### Author Rebuttal · Authors · 2025-07-29
>
> We sincerely thank you for your constructive comments. We are encouraged that you found our method both novel and practically valuable, and appreciate your recognition of its performance and efficiency improvements. We would like to address the comments and questions below.
>
> > W1: Labelling Error.
>
> We thank the reviewer for pointing this out. We will eliminate this error in the revised version to accurately reflect the intended operations within the resampler module.
>
> > W2: Evaluation task clarification.
>
> We thank the reviewer for pointing this out. We confirm that the evaluated task throughout the paper is indeed image classification, and we will explicitly clarify this in the revised version to avoid any ambiguity.
>
> > W3: Ambiguity in model usage and generalization claims.
>
> We thank the reviewer for raising this important concern. To clarify:
>
> 1. **Training vs. Evaluation Models:**
>    If we understand the question correctly, *training* refers to the models used to optimize the distilled dataset, while *evaluation* refers to the models trained on the distilled data to assess generalization. We have already provided this information in the original submission:
>    - For single-model distillation (FADRM), the training model is stated in the caption of **Table 1**.
>    - For multi-model distillation (FADRM+), the training models are detailed in **Table 7** of the Appendix.
>    - The evaluation models used are listed in the **top row of Table 1**.
>
> We **fully agree that these details are crucial** to the clarity and reproducibility of our method. To avoid any ambiguity, we will consolidate this information and explicitly state it in the main paper.
>
> 2. **Generalization Beyond ResNet:**
>    Our distilled datasets are designed to generalize beyond the ResNet family. As shown in the **right block of Table 2** in the main paper, we have already conducted cross-architecture evaluations to demonstrate that our method generalizes beyond the ResNet family. To further strengthen this point, we also include **additional results** in the table (ImageNet-1k IPC=50) below to more comprehensively support the cross-architecture generalizability of our approach.
>
> |  Models         | RDED | FADRM  | EDC | FADRM+ |
> |:-----------------:|:------:|:--------:|:-----:|:--------:|
> |  DeiT-Tiny      | 44.5 |**46.3**| 55.0|**55.3**|
> |  ConvNext-Tiny  | 65.4 |**66.8**| 66.6|**67.4**|
> |  Swin-Tiny      | 56.9 |**58.2**| 63.3|**64.7**|
> |  MobileNet-V2   | 53.9 |**54.1**| 57.8|**58.6**|
> |  ShuffleNet-V2  | 30.9 |**31.2**| 45.7|**46.3**|
> |  EfficientNet-B0| 57.6 |**58.4**| 60.9|**61.9**|
>
>
> We appreciate the reviewer’s thoughtful feedback and will revise the manuscript accordingly to enhance clarity and completeness.
>
> > W4: Clarification for Figure 2.
>
> We thank the reviewer for raising this point. We would like to clarify that in our framework, the original data patches shown in Figure 2 are not meant to represent the full original dataset, but rather a subset **selected as initialization** for the distilled dataset. Specifically, for a target of *N* images per class (e.g., 50 IPC), we first select *N* patches per class from the original dataset, which are then used as the initialization for optimization. Therefore, the number of original and distilled patches is intentionally the same at this stage. To avoid confusion, we will revise the caption of Figure 2 to better explain this initialization process.
>
> > W5: Inconsistent memory results.
>
> We sincerely thank the reviewer for pointing this out. Upon careful re-examination, we confirm that this was indeed a reporting error, the correct peak GPU memory usage should be 11.0 GB. We apologize for the oversight. Importantly, this correction does not affect any of the experimental conclusions or comparisons in the paper. We will update the result accordingly in the revision.
>
> > Q1: Performance evaluation task.
>
> We thank the reviewer for the question. Yes, the downstream task used for performance evaluation throughout the paper is **image classification**.
>
> This task was chosen because, given only class labels and distilled images, image classification provides a standardized way to assess whether the synthetic data captures sufficient discriminative information for training a performant classifier. It is also the **standard evaluation protocol** in the dataset distillation literature, particularly for the image modality. Therefore, it provides a fair and widely accepted basis for comparison across methods.
>
> > Q2: Trained Models and hyper-parameter configurations.
>
> We appreciate the reviewer’s question.
>
> In our main experiments (Table 1), we trained **ResNet-18**, **ResNet-50**, and **ResNet-101** on the distilled datasets. To further evaluate the generalizability of our method across architectures, we conducted **cross-architecture experiments (Table 2 right)** involving:
> - **Transformer-based models**: Swin-Tiny, ConvNext-Tiny, DeiT-Tiny
> - **Lightweight CNNs**: MobileNetV2, ShuffleNetV2
> - **High-capacity models**: EfficientNet, DenseNet201, Wide ResNet50-2
>
> All models were trained following the configuration used in EDC: 1000 epochs for CIFAR-100 and Tiny-ImageNet IPC=1, and 300 epochs for all other datasets, to ensure consistency and fair comparison across methods. Other hyperparameters (e.g., learning rate, optimizer) were selected based on model capacity and the complexity of the distilled dataset. Detailed training configurations are provided in Appendix Section F.

---

> > ### Comment · Reviewer_Ujgz · 2025-08-08
> >
> > My concerns have been addressed by the authors through rebuttal and I don't have further questions for discussion.

---

> > > ### Author Response · Authors · 2025-08-08
> > >
> > > We are very pleased that we were able to address your concerns and sincerely thank you for your thoughtful feedback. We would be most grateful if you might kindly consider raising your score. Thank you again for your time and consideration.

---

### Official Review · Reviewer_ccAs · 2025-07-03

**Clarity:** 2
**Significance:** 3
**Originality:** 3
**Rating:** 4
**Confidence:** 3

**Summary:**

This paper introduces FADRM (Fast and Accurate Data Residual Matching), a novel dataset distillation framework that extends residual connections to the data level to address the issue of information vanishing in synthetic data. The method uses data-level skip connections and multi-resolution optimization (MRO) to preserve both local and global information while significantly reducing memory and time costs. By injecting residuals from the original patches during the iterative optimization process, FADRM ensures better feature retention and faster convergence. Empirical results across CIFAR-100, Tiny-ImageNet, and ImageNet-1K show FADRM's advantage in accuracy, efficiency, and cross-architecture generalization.

**Questions:**

How sensitive is the method to the choice of merge ratio and number of residual injections, and could these parameters be learned or adapted during training?

**Ethical Concerns:**

["NO or VERY MINOR ethics concerns only"]

**Final Justification:**

Thanks for the authors' response, and they have addressed most of my concerns - I've raised the score accordingly.

**Limitations:**

Yes

**Quality:**

2

**Strengths And Weaknesses:**

S1: The core idea of extending residual connections from model-level to data-level is novel. It aims to address the information vanishing issue common in uni-level dataset distillation.

S2: The proposed Multi-Resolution Optimization (MRO) and Mixed Precision Training (MPT) reduce computational costs and makes the method scalable to large datasets like ImageNet-1K.

S3: The method consistently outperforms prior SOTA on several benchmarks across both accuracy and efficiency, with ablations and cross-architecture generalization evaluations.

W1: The method requires pretrained feature extractors (with fixed BatchNorm) to work. It’s unclear how robust the method is if the pretrained model is weak or mismatched to the downstream task.

W2: The theorems give motivation and justification for ARC (and MRO), but they aren't tightly integrated into the method design or used to guide any decisions so it reads a bit post-hoc. It would help if the authors can discuss how to make ARC adaptive or grounded the merge ratio in theory or training signals.

W3: The paper doesn't discuss where FADRM fails or when ARC/MRO might hurt performance. This makes the method look overfitted to certain hyperparameters.

W4: The paper talks a lot about information vanishing and density but lacks a well-defined, generalizable metric or visualization to evaluate it quantitatively.

---

> ### Author Rebuttal · Authors · 2025-07-28
>
> We sincerely thank you for your constructive comments. We are encouraged that you find our work novel with strong empirical performance on both efficiency and effectiveness. We would like to address the comments and questions below.
>
> > W1: Concern if weak or mismatched, distillation may degrade.
>
> We thank the reviewer for the insightful comment.
>
> As shown in Theorem 1, FADRM, like other uni-level distillation methods, is most effective when the pretrained model not too confident (relative weak), i.e., when its output has high entropy. In this regime, the mutual information upper bound is looser, allowing distilled data to better capture information from the original dataset. This is supported by both theory (Theorem 1) and experiments (see table below).
>
> |  Squeezed Epochs | Cross-Entropy   | FADRM Accurracy (%) |
> |:----------------:|:---------------:|:---------------:|
> |  50              |      0.0338     |         61.5    |
> |  100             |      0.0054     |         59.6    |
> |  150             |      0.0031     |         58.7    |
> |  200             |      0.0028     |         58.3    |
>
> Regarding architectural mismatch, our original submission includes cross-architecture results (Table 2, right), where FADRM consistently outperforms baselines. We further reinforce this with additional experiments in the table below on ImageNet-1k IPC=50.
>
> |  Models         | RDED | FADRM  | EDC | FADRM+ |
> |:---------------:|:----:|:------:|:---:|:------:|
> |  DeiT-Tiny      | 44.5 |**46.3**| 55.0|**55.3**|
> |  ConvNext-Tiny  | 65.4 |**66.8**| 66.6|**67.4**|
> |  Swin-Tiny      | 56.9 |**58.2**| 63.3|**64.7**|
> |  MobileNet-V2   | 53.9 |**54.1**| 57.8|**58.6**|
> |  ShuffleNet-V2  | 30.9 |**31.2**| 45.7|**46.3**|
> |  EfficientNet-B0| 57.6 |**58.4**| 60.9|**61.9**|
>
>
> > W2: Ground the merge ratio in theoretical anlysis.
>
> We thank the reviewer for the question. To ground the choice of merge ratio $\alpha$ in theory, we define the generalization bound of FADRM as:
>
> $$
> D(\alpha) =  \widehat{\mathcal{L}}\_{\text{FADRM}} + 2B \cdot \mathfrak{R}\_n(\mathcal{H} \circ \tilde{\mathcal{C}}\_{\text{FADRM}})
> $$
>
> Assuming the loss is $\lambda$-strongly convex and letting $\Delta := \frac{1}{n} \sum\_i \|\tilde{x}\_i^{\text{res}} - x\_i\|$, we have:
>
> $$
> \widehat{\mathcal{L}}\_{\text{FADRM}} \le \alpha \widehat{\mathcal{L}}\_{\text{res}} + (1 - \alpha) \widehat{\mathcal{L}}\_{\mathcal{O}} - \frac{\lambda}{2} \alpha(1 - \alpha)\Delta
> $$
>
> and, as shown in Appendix A.3:
>
> $$
> \mathfrak{R}\_n(\mathcal{H} \circ \tilde{\mathcal{C}}\_{\text{FADRM}}) \le \alpha \mathfrak{R}\_n(\mathcal{H} \circ \tilde{\mathcal{C}}\_{\text{res}}) + (1 - \alpha) \mathfrak{R}\_n(\mathcal{H} \circ \mathcal{O}) + L\_h \cdot \alpha(1 - \alpha)\Delta.
> $$
>
> Substituting both yields:
>
> $$
> D(\alpha) \le X\alpha + Y(1 - \alpha) + Z(\alpha - 1)\alpha,
> $$
>
> where:
> - $X := \widehat{\mathcal{L}}\_{\text{res}} + 2B \cdot \mathfrak{R}\_n(\mathcal{H} \circ \tilde{\mathcal{C}}\_{\text{res}})$
> - $Y := \widehat{\mathcal{L}}\_{\mathcal{O}} + 2B \cdot \mathfrak{R}\_n(\mathcal{H} \circ \mathcal{O})$
> - $Z := \frac{\lambda}{2} \Delta - 2B L_h \Delta$
>
> Minimizing this quadratic yields the optimal merge ratio:
>
> $$
> \alpha^\star = \frac{1}{2} + \frac{Y - X}{2Z}.
> $$
>
> Since in practice we use temperature-scaled KL divergence, the strong convexity constant satisfies $\lambda \approx 1 + (C - 1)\exp(-\frac{d}{T})$, which approaches the number of classes $C$ as the temperature $T$ increases. Consequently, the bias term $\frac{Y - X}{2Z}$ becomes negligible relative to the dominant $\lambda$ term, and the optimal merge ratio $\alpha^\star$ remains close to 0.5.
>
> This aligns with our empirical results in Table 4, where $\alpha = 0.5$ performs consistently well across datasets. Thus, our selection is both theoretically grounded and empirically validated.
>
>
> > W3: Discussion where FADRM fails.
>
> We appreciate the reviewer’s comment and would like to clarify that the potential failure modes of FADRM to certain hyperparameters are indeed discussed:
>
> - MRO (lines 240–243): We show that applying MRO without ARC leads to performance degradation, as the reduced resolution impairs the representational fidelity without the residual correction.
> - Merge ratio $\alpha$ (lines 243–248): We demonstrate that a large $\alpha$ leads to excessive reliance on the optimized data, reducing robustness to overfitting.
> - Residual frequency $k$ (lines 262–267): We observe that overly frequent residual injection introduces redundancy and leaves insufficient iterations per stage, leading to degraded synthetic images.
>
> We agree these points deserve further emphasis and we will include an explicit subsection in the revision to summarize failure case. Thank you again for pointing this out.
>
> > W4: Lacks a clear metric to quantify information vanishing.
>
> We appreciate the reviewer’s suggestion. To better quantify the issue of information vanishing, we report the generalization accuracy of models trained on distilled images across varying optimization steps (500–4000 iterations). As shown in the table below, prolonged optimization without ARC leads to performance degradation, indicating the issue of information vanishing. In contrast, applying ARC enables continued performance gains, demonstrating its ability to extract additional information during extended optimization. We will include and visualize this metric in the revision.
>
> | Iterations | FADRM w/ ARC (%) | FADRM w/o ARC Accuracy (%) |
> |:----------:|:------------------:|:--------------------:|
> | 500        |    40.0            |    39.7              |
> | 1000       |    45.2            |    44.8              |
> | 1500       |    47.2            |    46.5              |
> | 2000       |    47.7            |    46.1              |
> | 2500       |    48.0            |    46.5              |
> | 3000       |    48.0            |    45.4              |
> | 3500       |    48.2            |    44.5              |
> | 4000       |    48.4            |    43.1              |
>
>
> > Q1 (1): Sensitiveness of the Hyper-parameter.
>
> We thank the reviewer for the question. We clarify that FADRM is robust to both the merge ratio $\alpha$ and the number of residual injections $k$.
>
> For $\alpha$, our theoretical analysis (see response to W2) suggests that the optimal value lies near 0.5. Empirically (Table 4), performance remains stable around this value, with less than 0.7% variation. Notably, $\alpha = 0.5$ achieves the best result on ImageNet-1K, and consistent trends are observed on CIFAR-100 (see table below), where $\alpha = 0.5$ also yields the strongest performance.
>
> |  $\alpha$        | FADRM Accurracy (%)|
> |:----------------:|:---------------:|
> |  0.9             |       58.0      |
> |  0.8             |       58.9      |
> |  0.7             |       58.7      |
> |  0.6             |       59.6      |
> |  0.5             |     **61.5**    |
> |  0.4             |       59.2      |
> |  0.3             |       58.6      |
>
>
> For $k$, moderate values (e.g., 2–4) balance optimization depth and information retention. Performance varies within 0.2% (ImageNet-1K) and 1% (CIFAR-100) as shown in left panel of table 5, with $k{=}3$ performs the best across diferent datasets.
>
> Overall, FADRM requires minimal tuning and $k=3$ and $\alpha=0.5$ performs stably across different datasets.
>
>
> > Q1 (2): Adaptive Hyper-parameters during training.
>
> We thank the reviewer for this insightful question. While our method supports adaptive adjustment of hyperparameters, we emphasize that the default configuration, merge ratio $\alpha = 0.5$ and number of residual injections $k = 3$, already yields strong and stable performance across diverse settings.
>
> As established in our theoretical analysis, the optimal merge ratio $\alpha$ lies near 0.5. The remaining two hyperparameters are the downsampled input resolution $D\_{\text{ds}}$ and the number of residual injections $k$, which determines when residual information from the original dataset is injected.
>
> To avoid *information vanishing* caused by prolonged optimization in each stage, we inject residuals only when the current synthetic image is deemed converged. We adopt the following convergence criterion:
>
> - Cross-entropy loss $\mathcal{L}\_{\text{CE}} < 0.01$
> - BatchNorm regularization $\mathcal{L}\_{\text{BN}} < 60$
>
> To adaptively determine $D\_{\text{ds}}$, we use total variation (TV) to quantify image smoothness. Smoother images can be optimized at lower resolutions without loss of fidelity. Specifically, for a batch of images $x \in \mathbb{R}^{B \times C \times H \times W}$:
>
> $$
> \text{TV}(x) = \frac{1}{B} \sum\_{i=1}^{B} \left( \| x\_i[:, 1:, :] - x\_i[:, :-1, :] \|_1 + \| x\_i[:, :, 1:] - x\_i[:, :, :-1] \|\_1 \right)
> $$
>
> The downsampling factor is then computed as:
>
> $$
> \text{ShrinkRatio} = \min\left(0.8, \frac{0.5}{\max(\text{TV}, 10^{-4})} \right), \quad D\_{\text{ds}} = \max\left( \lfloor D \cdot \text{ShrinkRatio} \rfloor,\ 128 \right)
> $$
>
> As shown in the experiments blow on IPC=10, this procedure achieves performance comparable to manually tuned configurations. We will include these results in the revision. We agree this is a promising direction and sincerely thank the reviewer for highlighting it.
>
> |                  | Manually Selected |      Adaptive    |
> |:----------------:|:-----------------:|:----------------:|
> |  ImageNet-1k     |       47.7 %      |      47.5 %      |
> |  CIFAR-100       |       61.5 %      |      61.2 %      |

---

### Official Review · Reviewer_xpuv · 2025-07-03

**Clarity:** 3
**Significance:** 3
**Originality:** 2
**Rating:** 5
**Confidence:** 4

**Summary:**

This research theoretically points out that the mutual information between original dataset and synthetic dataset generated by uni-level decoupled dataset distillation methods has upper bound. To mitigate this problem, this research suggests the FADRM framework, which leverages the data-level skip connection concept. FADRM framework has two components: Multi-resolution optimization (MRO) to enhance efficiency and Adjustable residual connection (ARC) to mitigate the information vanishing. Through the diverse experimental results, FADRM shows the efficiency and effectiveness improvement.

**Questions:**

I like the paper overall and would be happy to raise my score if the authors address the few weaknesses and questions.

1. Is there any reason why $\tilde{x}$ should be optimized in the form of Cross-entropy + BN matching in Theorem 1? I also checked the proof in the appendix, but that condition is not used. I wonder if $f_\theta$ trained and frozen with $\mathcal{O}$ is sufficient to use the data processing inequality.

2. Any further clarification on Theorem 2 would be appreciated. There is no sufficient interpretation of Theorem 2, while it contains many statements.

3. I am curious about the advantages of MRO compared to synthetic image parameterization methods [1,2,3] and whether the FADRM framework can be used with synthetic image parameterization. I speculate that the synthetic image parameterization method might bypass the problem of carefully tuning the down-sampled size $D_{ds}$ of MRO as claimed by the authors. In particular, DDiF [4], with its resolution-agnostic nature, can enable a dynamically adjusted $D_{ds}$ rather than a static predefined $D_{ds}$.

[1] Dataset Condensation via Efficient Synthetic-Data Parameterization

[2] Remember the Past: Distilling Datasets into Addressable Memories for Neural Networks

[3] Dataset Distillation via Factorization

[4] Distilling Dataset into Neural Field

**Ethical Concerns:**

["NO or VERY MINOR ethics concerns only"]

**Final Justification:**

This paper introduces a new problem in previous studies, information vanishing, with theoretical analysis and proposes a simple but effective methodology to alleviate it. I requested a further theoretical analysis and expandability of the proposed methodology, as well as additional explanations, and the authors' responses addressed most of my concerns. Therefore, I believe the paper is good enough to be accepted.

**Limitations:**

See weaknesses and questions

**Paper Formatting Concerns:**

No paper formatting concerns.

**Quality:**

4

**Strengths And Weaknesses:**

**Strengths**
1. The theoretical analysis of the uni-level decoupled framework in Theorem 1 is impressive because it has an intuitive interpretation and clearly shows the limitations of uni-level methods.

2. The basic concept of FADRM, Data-level skip connection is a novel idea that has not been proposed.

3. This research has conducted many detailed experiments to prove the effectiveness of the proposed idea and shows high efficiency and performance improvement.

**Weaknesses**
1. Although this research have theoretically analyzed the limitations of the uni-level decoupled framework, the authors have not theoretically demonstrated the effectiveness of the proposed methodology FADRM in the same framework.

2. (Minor) The simple application of Mixed Precision Training (MPT) for the sake of efficiency falls short in terms of originality and contribution.

3. (Minor) If I understand correctly, budget $b$ or $\mathcal{B}$ refers to the optimization steps. However, in dataset distillation, budget usually refers to storage budget in relation to IPC, so this usage can be confusing and I suggest replacing it with the appropriate word.

---

> ### Author Rebuttal · Authors · 2025-07-29
>
> We sincerely thank you for your constructive comments. We are encouraged that you find our work novel with impressive theoretical insights and strong empirical validation. We would like to address the comments and questions below.
>
>
> > W1. While the limitations of the uni-level decoupled framework are discussed, the theoretical effectiveness of FADRM within this framework remains unproven.
>
> Thanks for raising this point. We will formally analyze the impact of FADRM, particularly ARC to mitigate the limitations of the uni-level framework.
>
> In FADRM's setting, $\hat{x}$ is generated by using both $f\_\theta$ and real image patch from $\mathcal{O}$ by iteratively fusing the real image patch to the intermediate optimized image. This introduces a new direct dependency:
>
> $$
> \mathcal{O} \rightarrow \hat{x} \leftarrow f\_\theta
> $$
>
> This breaks the conditional independence:
>
> $$
> \hat{x} \text{ is not conditionally independent of } \mathcal{O} \text{ given } f_\theta
> $$
>
> Thus, the Markov structure assumed in Data Processing Inequality no longer applies. By the chain rule:
>
> $$
> I(\hat{x}; \mathcal{O}) = I(f\_\theta; \hat{x}) + I(\hat{x}; \mathcal{O} \mid f\_\theta)
> $$
>
> - In the uni-level framework:
>
> $$
> I(\hat{x}; \mathcal{O} \mid f\_\theta) = 0
> $$
>
> - In our FADRM setting, the second term is strictly positive:
>
> $$
> I(\hat{x}; \mathcal{O} \mid f\_\theta) > 0
> $$
>
> Hence the mutual information is strictly increased in our FADRM settings.
>
>
> > W2. The use of Mixed Precision Training (MPT) primarily for efficiency lacks sufficient novelty.
>
> We thank the reviewer for asking this question. We would like to clarify that our contribution lies not in the use of MPT itself, but in introducing optimization-level efficiency into dataset distillation, an area where most prior work focuses on architectural or initialization-level shortcuts (e.g., lighter backbones or real-image initialization). Our use of MPT is thus novel within this setting and highlights a promising direction for improving efficiency in future research.
>
> > W3. Misleading notation of "budget".
>
> Thank you for the suggestion. We will revise the terms $b$ and  $\mathcal{B}$ to $n\_{\text{iter}}$ and $N\_{\text{iter}}$, respectively, to improve clarity and avoid confusion.
>
>
> > Q1. Loss condition not used in the proof.
>
> We thank the reviewer for raising this concern. The specific loss function (cross-entropy + BN matching) is not required for the proof; it is included to reflect typical objectives used in uni-level frameworks. Theorem 1 remains valid regardless of the loss formulation, as long as the original dataset $\mathcal{O}$ is not involved and the pretrained model is kept frozen during optimization, following directly from the data processing inequality.
>
> > Q2. Further clarification on Theorem 2.
>
> We appreciate the reviewer’s request for further clarification on Theorem 2. Below we provide an intuitive and formal explanation, focusing in particular on Equation 8.
>
> Theorem 2 aims to theoretically justify the generalization advantage of FADRM over training with uni-level framework's distilled data alone. Specifically, Equation 8 establishes a complexity gap between the original dataset $\mathcal{O}$ and the uni-level framework's distilled data $\tilde{\mathcal{C}}^{\text{res}}$:
>
> $$
> \mathfrak{R}\_n(\mathcal{H} \circ \mathcal{O}) - \mathfrak{R}\_n(\mathcal{H} \circ \tilde{\mathcal{C}}^{\text{res}}) < -\frac{L\_h \Delta (L\_l + 2B\alpha)}{2B}
> $$
>
> This inequality reflects the assumption that uni-level framework's distilled samples $\tilde{\mathcal{C}}^{\text{res}}$ being directly constructed to match the behavior of the target model, are easier to fit and therefore induce **higher Rademacher complexity** than orginal images. In contrast, $\mathcal{O}$ contains richer semantic variability, which naturally regularizes overfitting, resulting in a lower complexity.
>
> Theorem 2 then shows that **combining uni-level framework with the structured real data** leads to a better generalization bound. Under the assumption, the generalization bound of $h\_{\mathrm{FADRM}}$ is rigorously shown to be tighter than that of $h\_{\mathrm{res}}$, i.e.,
>
> $$
> \widehat{\mathcal{L}}\_{\mathrm{FADRM}} + 2B \cdot \mathfrak{R}\_n(\mathcal{H} \circ \tilde{\mathcal{C}}\_{\mathrm{FADRM}})
> < \widehat{\mathcal{L}}\_{\mathrm{res}} + 2B \cdot \mathfrak{R}\_n(\mathcal{H} \circ \tilde{\mathcal{C}}^{\text{res}}).
> $$
>
> The insight of this theorem is that, when synthetic data is highly optimized and thus induces greater hypothesis complexity, combining it with the more structured and regular original data can lead to a tighter generalization bound, accounting for both empirical risk and theoretical analysis.
>
>
> > Q3 (1): Comparison betwwen image parameterization and MRO.
>
> We thank the reviewer for this insightful question. **Multiple Resolution Optimization (MRO)** is a *training efficiency technique* that reduces the computational cost of pixel-level optimization by periodically operating at lower resolutions. In contrast, **image parameterization** replaces pixel optimization with a latent code and a shared decoder, aiming to reduce parameter count and improve representational compactness.
>
> Key advantages of MRO over image parameterization include:
>
> - **Lower Hyperparameter Sensitivity**: MRO introduces only a single downsampling factor $D_{\text{ds}}$, while parameterized methods require careful tuning of latent size, decoder architecture, and regularization.
>
> - **Preservation of Fine Details**: MRO retains full-resolution pixel expressiveness, which is crucial for high-resolution datasets like ImageNet.
>
> - **Simplicity and Broad Compatibility**: MRO is easy to implement, requires no auxiliary modules, and can be seamlessly integrated into any optimization-based distillation framework.
>
>
> > Q3 (2) Apply image parameterization within FADRM Framework.
>
> We thank the reviewer for this insightful question. **FADRM is compatible with image parameterization techniques such as HaBa, etc**. In this setting, instead of initializing pixel-space learnable synthetic images, we initialize trainable latent bases and trainable hallucinator networks. During training, images are decoded via the hallucinators, and the FADRM loss is computed on the decoded outputs. Gradients are backpropagated to update both components. Residual feature injections can still be applied periodically (e.g., every 500 iterations) to enhance information retention. Additionally, MRO can be integrated by performing forward passes at lower resolutions to reduce training cost.
>
> We appreciate the reviewer’s suggestion and believe that combining image parameterization with uni-level frameworks opens up promising directions for future research.
>
>
> > Q3 (3) Adaptive $D_{\text{ds}}$.
>
> We thank the reviewer for the thoughtful question. While DDiF is resolution-agnostic during generation, it still requires a fixed output resolution and cannot dynamically select the optimal scale for each image. Therefore, it does not directly help determine the downsampling factor $D_{\text{ds}}$ automatically in our setting.
>
> To avoid manually select $D_{\text{ds}}$, we use an adaptive strategy based on total variation (TV) to estimate image smoothness. Smoother images can tolerate greater downsampling without loss of fidelity. Given a batch $x \in \mathbb{R}^{B \times C \times H \times W}$:
>
> $$
> \text{TV}(x) = \frac{1}{B} \sum_{i=1}^{B} \left( \| x_i[:, 1:, :] - x_i[:, :-1, :] \|_1 + \| x_i[:, :, 1:] - x_i[:, :, :-1] \|_1 \right)
> $$
>
> We compute:
> $$
> \text{ShrinkRatio} = \min\left(0.8,\ \frac{0.5}{\max(\text{TV}, 10^{-4})} \right),\quad D_{\text{ds}} = \max\left( \lfloor D \cdot \text{ShrinkRatio} \rfloor,\ 128 \right)
> $$
>
> As shown below, this adaptive strategy achieves performance on par with manual tuning:
>
> |                  | Manually Selected |   Adaptive   |
> |:----------------:|:-----------------:|:------------:|
> |  ImageNet-1K     |      47.7%        |    47.5%     |
> |  CIFAR-100       |      61.5%        |    61.2%     |
>
> We will include these results in the revision and thank the reviewer for inspiring this direction.

---

> > ### Comment · Reviewer_xpuv · 2025-08-08
> >
> > I would like to thank the author for the detailed explanation, clarification, and new result. Most of my concerns have been resolved. I suggest that the authors include such additional information in the revised version. Hence, I tend to accept this paper.

---

### Note · Authors · 2025-08-12

Dear Area Chair and Reviewers,

We sincerely appreciate the reviewers' thoughtful feedback and valuable suggestions. In our rebuttal, we have addressed each point in detail, and we will revise the manuscript accordingly to incorporate all recommendations.

We are also encouraged by the positive remarks, such as "the core idea of extending residual connections from model-level to data-level is novel [**xpuv, ccAs, Ujgz, Ne6j**], "The proposed method is effective on both improving efficiency and performance." [**xpuv, ccAs, Ujgz, Ne6j, kJwL**], "The theoretical analysis is impressive" [**xpuv, Ne6j**]. We will reflect all comments and suggestions in our revision.

Regarding the novelty concern raised by Reviewer kJwL, we have provided a detailed clarification of our technical novelty in the rebuttal. We respectfully emphasize that, to the best of our knowledge, we are the first to introduce residual connections at the data level, and this design is further supported by a rigorous theoretical analysis that establishes formal guarantees for its benefits. The remaining reviewers reviewers have recognized this novelty, and kJwL acknowledged the conceptual novelty of our approach. We hope our detailed follow-up explanations have addressed this concern for this reviewer.

Best,

Authors

---

### Decision · Program_Chairs · 2025-09-17

**Decision:**

Accept (poster)

**Comment:**

This paper proposes Fast and Accurate Data residual Matching, a dataset distillation to address the information vanishing problem in synthetic data.

Most reviewers acknowledge the novelty and, after rebuttal, the experimental support provider; as such, they lean towards acceptance
One reviewer insists on the incremental nature of the approach, when it comes to optimization approaches. While individual models might be incremental, the rest of reviewers agree that the paper merits acceptance.